# The Effect of Leaf Plasticity on the Isolation of Apoplastic Fluid from Leaves of Tartary Buckwheat Plants Grown *In Vivo* and *In Vitro*

**DOI:** 10.3390/plants12234048

**Published:** 2023-11-30

**Authors:** Natalya I. Rumyantseva, Alfia I. Valieva, Yulia A. Kostyukova, Marina V. Ageeva

**Affiliations:** 1Kazan Institute of Biochemistry and Biophysics, FRC Kazan Scientific Center of RAS, Lobachevsky str., 2/31, Kazan 420111, Russia; valievaalfiya@mail.ru (A.I.V.); j.kostyukova@mail.ru (Y.A.K.); mageeva58@mail.ru (M.V.A.); 2Department of Botany and Plant Physiology, Institute of Fundamental Medicine and Biology, Kazan Federal University, Kremlyovskaya 18, Kazan 420008, Russia

**Keywords:** *Fagopyrum tataricum*, micropropagated plants, apoplast washing fluid, low lighting, high relative humidity, leaf traits, stomata, cuticle, phenotypical plasticity, adaptation

## Abstract

Vacuum infiltration–centrifugation (VIC) is the most reproducible technique for the isolation of apoplast washing fluid (AWF) from leaves, but its effectiveness depends on the infiltration–centrifugation conditions and the anatomical and physiological peculiarities of leaves. This study aimed to elaborate an optimal procedure for AWF isolation from the leaves of Tartary buckwheat grown in *in vivo* and *in vitro* conditions and reveal the leaf anatomical and physiological traits that could contribute to the effectiveness of AWF isolation. Here, it was demonstrated that leaves of buckwheat plants grown *in vitro* could be easier infiltrated, were less sensitive to higher forces of centrifugation (900× *g* and 1500× *g*), and produced more AWF yield and apoplastic protein content than *in vivo* leaves at the same forces of centrifugation (600× *g* and 900× *g*). The extensive study of the morphological, anatomical, and ultrastructural characteristics of buckwheat leaves grown in different conditions revealed that *in vitro* leaves exhibited significant plasticity in a number of interconnected morphological, anatomical, and physiological features, generally driven by high RH and low lighting; some of them, such as the reduced thickness and increased permeability of the cuticle of the epidermal cells, large intercellular spaces, increase in the size of stomata and in the area of stomatal pores, higher stomata index, drop in density, and area of calcium oxalate druses, are beneficial to the effectiveness of VIC. The size of stomata pores, which were almost twice as large in *in vitro* leaves as those in *in vivo* ones, was the main factor contributing to the isolation of AWF free of chlorophyll contamination. The opening of stomata pores by artificially created humid conditions reduced damage to the *in vivo* leaves and improved the VIC of them. For *Fagopyrum* species, this is the first study to develop a VIC technique for AWF isolation from leaves.

## 1. Introduction

The plant apoplast is most commonly defined as the continuum of the cell walls and the intercellular spaces filled by air and apoplastic fluid [1]. Apoplast plays a crucial role in mineral nutrition [2], defense [3,4,5,6], plant development [7,8], and senescence [9]. Apoplastic fluid contains ions [1,2,10,11], reactive oxygen species [12,13], secondary metabolites [14], lipids [15], ascorbate [7,16,17], glutamate [18], peptides [4,19], proteins [1,3,20], diverse species of sRNAs [21,22], and perhaps a much wider variety of substances than we currently assume. 

Proteins enter the apoplast through the classical and non-classical routes of secretion and perform multiple functions in the apoplast: they are involved in cell wall synthesis and modifications [20,23,24], intercellular interaction [1], signal transduction [1,4,18], the maintenance of cellular redox status [13,17], the regulation of cell growth, differentiation [7,17], morphogenesis [8], and plant immune responses [5]. Thus, the apoplast is located “at the crossroads” of different directions of the biological sciences, and its comprehensive study needs multidisciplinary approaches combined with novel tools and methods. 

Despite the fact that plant secretomics has made a tremendous step forward over the past ten years [20,25,26,27], secretome studies are lacking for many agriculturally and pharmaceutically important plants. To some extent, it can be explained by the poorly developed methodology of apoplastic protein isolation for these plant species. To date, several methods for the isolation of apoplastic fluid have been elaborated, but the most well-established and reproducible for different plant species remains the vacuum infiltration–centrifugation (VIC) method due to its efficiency and simplicity [1,9,28,29]. In the first stage of VIC, the air of the intercellular space is replaced by a buffer, and the undissolved substances of apoplast are solubilized in the buffer. In the second stage, the fluid from intercellular spaces is collected by centrifugation; this fluid is referred to as either apoplastic fluid [1,20], apoplastic washing fluid (AWF) [10,11], or intercellular (washing) fluid [30,31]; the use of all terms is equivalent. The well-developed VIC technique allows for the extraction of apoplastic proteins without much cell damage [32].

Many factors can affect AWF yield and purity (free from contamination by intracellular proteins) of AWF [20]. In general, for each plant species, the VIC technique requires some refinement depending on the species-specific physiological and anatomical traits [10,20,31,33,34]. However, it should be kept in mind that the physiological or morphological traits affecting the VIC procedure are not completely species- and organ/tissue-specific but also depend on the age of the plant and the environmental conditions [29]. 

Plants have limited mobility and their survival depends on how promptly they can change their homeostasis to adapt to fluctuations in the external environment. Therefore, one of the key mechanisms by which plants survive and adapt to a rapidly changing climate is phenotypic plasticity, which can be defined as the ability of a single genotype to express different phenotypes in response to a change in its environment [35]. Based on this, the main forces shaping the structure and functions of plants can be considered to be environmental stresses, which exclude the reproduction of non-plastic forms and give an advantage to phenotypes undergoing rapid acclimatization to a variety of abiotic and biotic stressors. It is known that the parameters of plant anatomical structure and especially leaf structure (leaf sizes, stomata sizes, stomata density, trichome morphology, and cuticle thickness) are usually the most plastic characteristics and depend on such environmental factors as drought, light intensity, daily and nightly temperatures, humidity, exposure to pathogens, etc. [36].

*In vitro*-cultured plants (plantlets) demonstrate significant changes in their habitus, physiology, and anatomy (dwarfism, poor photosynthetic efficiency, a decrease in the epicuticular wax layer, the malfunctioning of stomata, and their enormous “openness”), often seen as disorders [37]. It can be assumed that the driving force behind these changes is stress caused by unusual environmental conditions, different from the natural conditions of plant growth [38]. These are relative humidity (RH) of about 100%, low light intensity and another light spectrum, heterotrophic nutrition, wounding during cutting, high sucrose content, and special mineral composition of the nutrient medium. Nevertheless, morphological and physiological ”abnormalities” unique to plants propagated *in vitro* can be considered rather as acclimatization to very particular environmental conditions since the majority of them do not interfere with high rates of plant micropropagation and disappear during ex vitro transfer and the subsequent growth of plants in field or greenhouse conditions. Clearly, if the success of the VIC technique depends on the density of stomata and the area of stomatal pores in leaves, then the quantity, quality, and purity of AWF may also depend on the range of plasticity of these traits, and *in vitro*-cultured plants could be favorable models for this study. 

According to numerous original reports, the secretome of any plant is a highly dynamic system composed of a mixture of two different protein populations, one of which is constitutively secreted into the apoplast and the other is secreted in response to environmental changes [1]. It can be expected that the secretomes of leaves *in vitro* and *in vivo* will vary significantly due to strong differences in environmental conditions for both objects. On the other hand, *in vitro* plants grown under a controlled environment will probably have a more reproducible combination of extracellular proteins compared to plants grown in field or greenhouse conditions. Moreover, some anatomical traits typical to various plant species micropropagated *in vitro* (a thin cuticle, a decrease in epicuticular wax, an “openness” of stomata) may be favorable for AWF isolation, whereas such traits inherent to *in vivo*-grown plants as semi-open stomata and a thick cuticle impregnated and covered with epicuticular wax [39,40] can impose restrictions on the efficiency of the VIC technique. However, because we are unfamiliar with research that compares the isolation of AWF from the leaves of plants cultured *in vitro* with plants grown *in vivo*, it is difficult to estimate how accurate our assumptions are. 

Tartary buckwheat (*Fagopyrum tataricum* (L.) Gaertn.), a pharmacologically and nutritionally valuable pseudocereal plant [41,42], belongs to the *Polygonaceae* family [43] and originates from the eastern side of the Tibetan Plateau [44]. It is a self-pollinator, unlike common buckwheat (*Fagopyrum esculentum* Moench), and has a small genome (489.3 Mb) that is RNA-sequenced [45] (http://www.mbkbase.org/Pinku1, accessed on 20 August 2018). Tartary buckwheat fruits are gluten-free and rich in metabolites that are beneficial to human health and have anti-oxidative, anti-cancer, anti-hypertension, anti-diabetic, cholesterol-lowering, and cognition-enhancing properties [46,47]. The most well-known of those whose positive effect on human health has been proven are the flavonoids rutin and quercetin. Buckwheat is the only plant among cereals and pseudocereals whose fruits accumulate rutin and its content is more than 100–200-fold higher in Tartary buckwheat than in common buckwheat [48,49]. *Fagopyrum tataricum* has 61 flavonoids and 94 non-flavonoid metabolites, with significantly higher levels than those found in common buckwheat [50]. The interest in buckwheat, namely, Tartary buckwheat, has increased significantly in the last five years; this was evident from reviews on its useful properties, biotechnology, and genetics [41,47,51,52,53]. Nevertheless, no studies have been performed on the secretomics of *Fagopyrum* species. In part, this may be due to the lack of methods and approaches for the isolation of apoplastic fluid for this object. 

This study aimed to elaborate an optimal procedure for AWF isolation from the leaves of Tartary buckwheat grown *in vivo* and *in vitro* and reveal the leaf anatomical and physiological traits that contribute to the effectiveness of AWF isolation. We showed for the first time that leaves of Tartary buckwheat plants grown *in vitro* exhibited significant plasticity in a number of interconnected morphological, anatomical, and physiological features, generally driven by high RH and low lighting; some of them, such as the increased permeability of the cuticle of the surficial cells, large intercellular spaces, increase in the size of stomata and in the area of stomatal pores, drop in density, and area of calcium oxalate druses, are beneficial to the effectiveness of the VIC method. For *Fagopyrum* species, this is the first study to develop a VIC procedure for isolating AWF from leaves.

## 2. Materials and Methods

### 2.1. Plant Material

Tartary buckwheat (*Fagopyrum tataricum* (L.) Gaertn.) plants were grown outdoors (hereinafter: *in vivo* plants) at the vegetation site of the Kazan Institute of Biochemistry and Biophysics KSC of RAS, Russia (55°47′33.86″ N, 49°7′15.62″ E; 85 m above sea level) and in aseptic conditions *in vitro* (hereinafter: *in vitro* plants). Seeds of Tartary buckwheat, sample k-17, were obtained from the collection of the N.I. Vavilov Institute of Plant Genetic Resources, Saint Petersburg, Russia, and sown in early May (May 1–10). Outdoor plants were grown in pots filled with the commercial soil “Veltorf” with daily watering. The vegetation site was protected from rainfall by a light- and UV- transparent polycarbonate canopy. The main experiments with plants were conducted in June of the years 2020–2022; the meteorological conditions for the experimental period (mean daily temperatures, relative humidity, number of sunny days) are presented in Appendix A. Photosynthetic photon flux density (PPFD) changed from 170–210 μmol m^−2^s^−1^ (cloudy days) to 600–700 μmol m^−2^s^−1^ (sunny days). In the studies, 3–4 cm wide leaves from 4–5-week-old plants which were on the stage before the inflorescence formation, were used. 

In order to obtain *in vitro* micropropagated plants, buckwheat seeds were dehulled from the pericarp and surface-sterilized in 70% (*v*/*v*) ethanol (2 min) followed by a 2% sodium hypochlorite solution for 10 min, rinsed thrice in distilled sterile water, and germinated on tap water solidified with 0.6% agar for 3 days in the dark at 25 °C. Then, the upper part of the seedlings was cut off 1 cm above the cotyledon leaves, transferred into jars containing an MS [54] basal salt medium (pH 5.5–5.6), and supplemented with 2.0 mg/L thiamine∙HCl, 1.0 mg/L pyridoxine∙HCl, 1.0 mg/L nicotinic acid, 100 mg/L myoinositol, 30 g/L sucrose, and 7 g/L agar. Plants were grown at 25 ± 2 °C under cool-white fluorescent tubes with a photosynthetic photon flux density (PPFD) of 40 μmol m^−2^s^−1^ over a 16 h photoperiod. The micropropagation cycle was a 30-day subculture of nodal segments of 2 cm in length on an MS medium without hormones. For all experiments, *in vitro* plants were approximately 28–30 days old. For the experiments for the study of the effect of CaCl_2_ concentration on CaOx druse formation, a double concentration of anhydrous CaCl_2_ was added to the MS medium and the subculture was 30 or 45 days old. 

### 2.2. Leaf Morphology Analysis

The length, width, and leaf blade area of one-month-old plants grown *in vitro* and *in vivo* were measured on 15 selected leaves using ImageJ software (version 1.53k Java 1.8.0_172) [55]. 

### 2.3. Determination of Leaf Dry Weight 

More than 100 leaves were used to determine the dry weight of the leaves. The collected fresh leaves were weighed and then oven-dried for 2 h at 60 °C until samples attained a constant weight. Then, the dry leaves were weighed, and the dry weight was expressed as a percentage of the fresh weight.

### 2.4. Light Microscopy and Transmission Electron Microscopy

Areas of *in vivo* and *in vitro* leaves that were used for histological and electron microscopy investigations are shown in Appendix A. For transmission electron microscopy (TEM), five tissue samples (3 × 5 mm) from the area of lateral leaf veins were fixed in 2.5% glutaraldehyde (Fluka, Buchs, Switzerland) in a 0.1 M phosphate buffer (pH 7.2), washed in the same buffer, and then post-fixed in 1% OsO_4_ (Alfa Aesar, Haverhill, MA, USA) diluted with the same buffer solution to which sucrose was added (25 mg/mL) for 3 h at room temperature. Tissues were dehydrated through a graded ethanol series, 100% acetone, and 100% propylene oxide and embedded in Epon-812 epoxy wax (Sigma-Aldrich Chemie, Buchs, Switzerland). Sections (70–100 nm thick) were cut on an “LKB 8800” (LKB, Stockholm, Sweden) ultramicrotome, collected onto nickel grids (Sigma-Aldrich, Grantham, UK), and stained with 4% uranyl acetate (Serva, Heidelberg, Germany) for 20 min and Reynold’s lead citrate for 10 min [56]. Observations were made using a “Hitachi 7800” transmission electron microscope (Hitachi, Tokyo, Japan). For the histological analysis, semithin sections of 3 µm thick were cut on an Ultra Cut E (Reichert-Jung, Vienna, Austria) ultramicrotome and stained with 0.5% toluidine blue (TB) (Sigma, St Louis, MO, USA) according to Trump’s method [57]. The stained sections were examined under a Jenamed microscope (Carl Zeiss, Potsdam, Germany) and recorded digitally using an AxioCam MRc5 camera with AxioVision Rel.4.6 software (version AxoiVs40 V 4.8.2.0) (Carl Zeiss MicroImaging GmbH, Jena, Germany).

### 2.5. Epidermal Peel Microscopy

The vital structure of adaxial and abaxial leaf epidermis was studied on single-layered epidermal peels. Areas of *in vivo* and *in vitro* leaves that were used for the morphological analysis of stomatal and epidermal cells are shown in Appendix A. The epidermis was manually peeled from both sides of a leaf with forceps and placed in a drop of distilled water for microscopy. For the investigations, 2–3 plants were used and three leaves from each plant were examined. The measurements were made on three epidermal peels from each side of the leaf.

The epidermal cell area (ECA), epidermal cell density (ECD, number of cells per mm^2^ of area), stomata sizes (length (SL) and width (SW), stomata pore sizes (stomatal pore length (SPL), stomatal pore width (SPW), stomatal pore area (SPA), and stomatal density (SD, number of stomata per mm^2^ of area) were measured on at least three preparations of adaxial and abaxial epidermal peels per leaf. In total, 50 measurements were made on one peel.

Stomatal index (SI) was calculated according to the following formula [58]:
(1)
$$SI(\%)=\frac{SD}{SD+ECD}\times 100,$$

where SI—stomatal index, SD—stomatal density, ECD—epidermal cell density.

CaOx druse study was carried out on preparations of epidermal peels taken off in the areas of first-order veins (Appendix A). For studying CaOx druses, the epidermis was taken off only from the abaxial side of the leaf.

The number of druses per mm^2^ of area (druse density) and the areas of druses were calculated on preparations of epidermal peels with veins and adjacent mesophyll cells.

In order to compare the frequency of occurrence of large and small CaO druses among different study objects, the data samples were normalized by converting them from 0 to 1 according to the following formula: a = (x − x_min_)/(x_max_ − x_min_),(2)
where x_min_ equals 0.

CaOx crystal solubilization was performed with 0.5 M Na-EDTA (pH 10.0) [59]. The chelating solution was added to the epidermal peel preparation that had previously been examined. The same procedure was applied to the precipitate formed by centrifugation of AWF at 10,000× *g*.

The vital epidermal peels were examined under a Jenamed microscope (Carl Zeiss, Germany), photographed by an AxioCam MRc5 digital camera (Carl Zeiss, Germany), and processed using Axiovision Rel.4.6 software (Carl Zeiss, Germany) and ImageJ software [55].

### 2.6. Toluidine Blue Test to Assay the Cuticular Permeability 

To detect defects in leaf cuticles, we used a TB test elaborated by Tanaka et al. (2004), with some modifications. An aqueous solution of 0.05% (*w*/*v*) TB (Sigma, St. Louis, MO, USA) was poured onto the Petri dishes, and leaves were submerged in the dye solution. Only the leaf blade was stained; the petiole remained above the dye solution’s surface. After certain intervals, starting from 2 min to 1 h, the leaves were washed in distilled water to remove excess TB and viewed under a stereo microscope (Optika Microscopes, Ponteranica, Italy), analyzing the staining of epidermal cells. Then, the details of staining were investigated microscopically on epidermal peels made from both sides of a leaf. Five leaves from each plant were examined. The measurements were made on three epidermal peels from every side of the leaf. The vital epidermal peels were examined under a Jenamed microscope (Carl Zeiss, Germany), photographed by an AxioCam MRc5 digital camera (Carl Zeiss, Germany), and processed using Axiovision Rel.4.6 software (Carl Zeiss, Germany).

### 2.7. Phenotypic Plasticity Index (PPI) Estimation

Phenotypic plasticity index (PPI) was calculated as described by Larcher et al. [60], using the following expression for each of the parameters: PPI = (maximum mean values − minimum mean values)/maximum mean value,(3)
where PPI ranges from zero to one, 0 indicates no plasticity, and 1 indicates the maximum plasticity. 

### 2.8. Apoplast Washing Fluid Isolation Procedure 

Approximately an hour before the harvesting of leaves, plants grown *in vivo* were watered. The total weight of the leaves sampled for the isolation of AWF was 0.4–0.7 g. Leaves of 3–4 cm width (Figure 1a) were detached from the plant at the petiole using a razor blade and put in a shallow basin with distilled water to remove any contaminants from the wounded surface. Afterwards, leaves were consecutively rinsed—first with a 0.1% Tween 20 solution and then with distilled water—and dried carefully by blotting them with absorbent paper. For leaf infiltration followed by AWF isolation, a soft plastic grid for packing goods, cut into 6 × 6 cm squares, was used. Leaves were placed on half of the grid surface, as shown in Figure 1a, and the overall weight of the leaves and the grid was measured. Then, the free end of the grid was folded so that the leaves were inside the grid bag, which was carefully rolled, put into a 20 mL plastic needleless syringe by directing leaf tips to the needle part of the syringe, and filled with a 15 mL Na-phosphate buffer (pH 6.5) containing proteinase inhibitors (10 μM phenylmethylsulfonyl fluoride (PSMF) (Sigma-Aldrich, St Louis, MO, USA), 0.5 mM Na-EDTA (Sigma-Aldrich, USA)) for the infiltration procedure. After removing air bubbles, the syringe tip was covered with a gloved finger, and negative pressure was created by pulling the plunger slowly down to a mark of 20 mL. The slow release of the plunger was accompanied by filling the airless intercellular spaces with a Na-phosphate buffer (Figure 1c). The infiltration step was repeated until the leaves were fully infiltrated, which was controlled visually by changing the leaf color from green to dark green. 

After infiltration, the plastic grid with leaves was removed from the syringe. The leaves were dried with absorbent paper and weighed without removing them from the plastic grid. To collect AWF, we used a system consisting of several plastic tubes and syringes of different sizes assembled as a “nesting cup”: a 2 mL syringe (which could be replaced by a 1 mL plastic tip); a 20 mL syringe trapped to a plastic Eppendorf tube with Parafilm; and a 50 mL Falcon centrifuge tube (Figure 1d). The grid with the leaves was wrapped around a 2 mL syringe so that the tips of the leaves were oriented to the needle part of the syringe (Figure 1e) and placed inside a 20 mL plastic syringe, which, in turn, was inserted into a Falcon tube (Figure 1e). To choose the optimal centrifugation rate, the infiltrated leaves were centrifuged for 10 min at 200× *g* or 600×/900×/1500×/3000× *g* in a swinging bucket centrifuge (Eppendorf AG, Hamburg, Germany). 

After centrifugation, the Eppendorf tube with the gathered AWF (Figure 1f) was detached from the syringe and centrifuged for 3 min at 10,000× *g* in a MiniSpin centrifuge (Eppendorf AG, Hamburg, Germany). The supernatant was removed to a new Eppendorf tube and used to estimate the AWF yield and protein content in it. 

Apoplast washing fluid yield was estimated as the weight of collected AWF (equated numerically to its volume) divided by the weight of leaves before infiltration (µL/g fresh leaf weight). 

To study the precipitate content after centrifugation, a drop of water was added to it, mixed, mounted on the glass slide, and viewed under a light microscope. To detect starch grains in precipitates after AWF centrifugation, Lugol’s staining was performed [61].

The main features of AWF isolation from leaves *in vitro* (Figure 1b) were as follows: 1. Cut leaves (1 cm × 1.5 cm) were immediately placed on a plastic grid in a wet chamber; 2. The procedure for the treatment of leaves with detergent was omitted; 3. Washing of wounding contaminants was carried out in a grid bag to reduce the effect of drying on the leaves. Other steps of AWF isolation were the same as those for *in vivo* leaves.

### 2.9. Protein Content 

Protein content was determined by the Bradford method [62].

### 2.10. Malate Dehydrogenase (MDH) Activity Assay 

Malate dehydrogenase activity was assessed by the NADH oxidation rate [63]. To 2.9 mL of a reaction mixture consisting of 0.13 mM β-NADH-Na_2_ (Sigma-Aldrich, USA) and 0.25 mM oxalate acetic acid (Acros Organics, Geel, Belgium) in a 0.1 M Na-phosphate buffer (pH 7.5), 100 μL AWF was added. The change in optical density was recorded at 25 °C for 180 s at 340 nm on a LAMBDA 25 spectrophotometer (Perkin Elmer, Shelton, ST, USA). 

### 2.11. Statistical Analyses and Software

The statistical analysis of the data was performed using SigmaPlot software (version 11.0). Data are presented as mean ± standard error. *t*-tests at *p* ≤ 0.05 were applied; statistically significant differences are indicated by different letters. PCA analysis was run on Origin software (version 9.0.0). 

## 3. Results

### 3.1. Elaboration of an Optimal Procedure for AWF Isolation from the Leaves of Buckwheat Plants Grown In Vivo and In Vitro

The stages of leaf preparation and infiltration had some distinctions for the leaves of plants grown *in vivo* and *in vitro*. For *in vitro* leaves, a single-step infiltration was sufficient to accomplish the complete saturation of leaves with a buffer within 30 s. For *in vivo* leaves, the duration of infiltration was prolonged to 2 min or more since the infiltration procedure had to be repeated 3–4 times to fill the intercellular spaces with the buffer. Rinsing the leaves of outdoor plants with a 0.1% Tween 20 solution as an additional step in the VIC method slightly increased the efficiency of the infiltration stage, reducing the number of vacuum applications to 2–3 repetitions.

An important detail applied to the device for the VIC procedure was a soft plastic 5 × 5 mm square grid. Unlike Parafilm and Miracloth, which have been used previously for the VIC method [14,64], the plastic grid did not prevent the free outflow of fluid during centrifugation, held the leaves around the central axis well, and protected them from deformation. In addition, when the leaves were in the grid, they were easy to manipulate: to get out of the syringe after the infiltration process, to dampen with a paper towel, and to transfer to the AWF-collecting apparatus. For small leaves of *in vitro*-grown plants, this is the only acceptable way to minimize damage.

Then, the effect of centrifugation force on AWF yield and protein content was investigated (Table 1). The infiltrated leaves were centrifuged at 200× *g*, 600× *g*, 900× *g*, 1500× *g*, and 3000× *g* for 10 min. In general, the procedure of centrifugation at all forces followed the rule “the higher the centrifugation force is, the greater the yield of AWF and protein content in it are”. As shown in Table 1, the yield of gathered AWF as well as AWF protein content increased with increasing centrifugation force (Table 1). It should be noted that the values of AWF yield and AWF protein content at 200× *g*, 1500× *g*, and 3000× *g* had a more dispersed dataset compared to 600× *g* and 900× *g* (Table 1).

The comparison of AWF yield and AWF protein content retrieved at different centrifugal forces showed that centrifugation at 200× *g* was ineffective both for *in vivo* and *in vitro* leaves (Table 1) because AWF yield and protein content were definitely low. For both leaf variants, an increase in centrifugal force to 600× *g* resulted in a substantial increase in the AWF yield and protein content compared to an increase to 200× *g*. It is important to note that at 600× *g*, both the AWF yield and protein content were much higher for *in vitro* than for *in vivo* leaves. At 900× *g*, the protein content of AWF was also higher for *in vitro* leaves than for *in vivo* leaves, but the differences between both variants were not statistically significant. Nevertheless, since the yield of AWF collected was higher for *in vitro* than for *in vivo* leaves, the total yield of apoplast proteins collected at 900× *g* was also higher for *in vitro* leaves compared to *in vivo* ones.

Noteworthily, there were no statistically significant differences in the AWF yield and apoplastic protein content for *in vitro* leaves centrifuged at 900× *g* and 1500× *g*, whereas, for *in vivo* leaves, AWF yield was significantly higher at 1500× *g* than at 900× *g*. At centrifugal forces of 1500× *g* and 3000× *g*, there were no statistically significant differences in the AWF yield and protein content in variants “*in vivo*” and “*in vitro*”, although the tendency to increase both the AWF yield and protein content was observed at 3000× *g* as compared to 1500× *g* in both variants. Eventually, it can be concluded that for *in vivo* and *in vitro* leaves, the same yield of AWF yield and protein content occurred at different centrifugal forces: at 900× *g* for *in vitro* leaves and 1500× *g* for *in vivo* leaves.

According to the AWF extraction procedure described in Methods, the AWF collected from infiltrated leaves was additionally centrifuged at 10,000× *g*. This approach was necessary to remove the remnants of destroyed leaf cells (organelles, cell wall fragments, etc.), if any. Typically, centrifugation at 10,000× *g* of *in vivo* AWF samples collected at 900× *g*, 1500× *g*, and 3000× *g* resulted in the appearance of a green precipitate in the form of plaque on the inner walls of the tube or in the form of sediment at the bottom of the tube (Figure 2a). The color of AWF after 10,000× *g* centrifugation also varied: from colorless in samples collected at 200× *g* and 600× *g* to yellowish in samples collected at 900× *g*, 1500× *g*, and 3000× *g* (Figure 2c). In contrast, *in vitro* AWF samples did not produce any visible precipitate at 10,000× *g* centrifugation (Figure 2b) and were colorless or had a slightly yellowish color at 3000× *g* (Figure 2d). 

Cytological analysis showed that the plaque after the centrifugation of leaves at 900× *g* was composed of chlorophyll clumps and, in rare cases, whole palisade mesophyll cells. Apparently, the green color of the plaque was due mainly to chlorophyll leakage from destroyed chloroplasts in mesophyll cells. After centrifugation of *in vivo* leaves at 1500× *g* and 3000× *g* and staining of the precipitate with Lugol’s reagent, we found chlorophyll clumps with randomly scattered starch granules, the whole and destroyed cells of palisade mesophyll, peltate trichomes, simple trichomes bearing numerous small CaOx prismatics, and idioblasts with one CaOx crystal (Appendix A). The number of trichomes and CaOx idioblasts was found to be increased at higher centrifugation forces (1500× *g*, 3000× *g*) and they were very rare at 600× *g* and 900× *g* modes of centrifugation. 

The degree of cytoplasmic contamination of AWF was assessed by measuring MDH activity (Figure 3). For *in vitro* leaves, MDH activity in AWF samples collected at 200× *g*, 600× *g*, and 900× *g* was mostly undetectable. For *in vivo* leaves, at the same centrifugation speeds, MDH activity was detected but was low. In AWF collected at 1500× *g* and 3000× *g* (*in vitro* leaves), MDH activity was insignificantly higher than at lower centrifugal forces. In contrast, in AWF collected from *in vivo* leaves, especially those centrifuged at 3000× *g*, MDH activity was significantly higher than at other centrifugation forces. Nevertheless, in all AWF samples, even those collected at 3000× *g*, MDH activity was not more than 1% of the cytoplasmic level.

Thus, methodological experiments on the selection of infiltration–centrifugation conditions demonstrated that the leaves of Tartary buckwheat plants grown *in vivo* and *in vitro* showed significant differences in buffer infiltration ability, sensitivity to the centrifugation mode, AWF yield, and apoplastic protein content. 

### 3.2. Morphological and Anatomical Traits of Leaves of Tartary Buckwheat Plants Grown In Vivo and In Vitro and Their Phenotypical Plasticity 

Since the efficiency of the VIC procedure depends to a great extent on the leaf anatomy [31], some morphological traits and the anatomical structure of the leaves of Tartary buckwheat plants grown *in vivo* and *in vitro* were studied. They varied in dry mass, thickness, length, width, and leaf blade area (Table 2). The sizes of *in vitro* leaves were significantly smaller than those of *in vivo* leaves (Figure 1a,b, Table 2). These parameters had the highest PPI: 0.69, 0.73, and 0.93 for length, width, and area of leaf blade, respectively (Table 2). In contrast, the thickness and dry mass of the leaf had much smaller differences between the *in vivo* and *in vitro* leaves (PPI: 0.14 and 0.23, respectively) (Table 2). A distinctive morphological feature of the *in vivo* leaves of Tartary buckwheat was the presence of a bright anthocyanin spot at the place where veins were branching out from the petiole and anthocyanin staining along the margin of the leaf blade (Figure 1a). In these places, anthocyanins were accumulated in the simple, non-secretory trichomes and epidermal cells. In the *in vitro* leaves, such patterns of anthocyanin localization were found very rarely (Figure 1a,b).

Histological analysis demonstrated that the leaves of *F. tataricum* grown in both environments were characterized by dorsiventral anatomy, with the epidermis showing an amphistomatic condition (stomata occurred on both the upper and the lower sides of the leaves) (Figure 4). *In vivo* and *in vitro* leaves had one-layered adaxial and abaxial epidermises, several rows of palisade mesophyll consisting of elongated cells with the peripherally located chloroplasts, a spongy mesophyll formed by rounded cells with chloroplasts, and vascular bundles (Figure 4a,b). The sizes of intercellular spaces between mesophyll cells and the sizes of sub-stomatal cavities were larger in the *in vitro* leaves compared to the *in vivo* leaves (Figure 4a,b). Moreover, in *in vitro* conditions, the palisade mesophyll generally had only one or two rows of cells instead of three or four in outdoor conditions (Figure 4a,b) and its sizes *in vitro* were almost 2-fold smaller than those *in vivo* (Figure 4a,b); the latter was especially evident in preparations of isolated mesophyll cells (Appendix A). 

A histological study showed that in both environments, some cells of the adaxial epidermis accumulated intravacuolar phenolics and proteins; phenolics turned gray or black after fixation with OsO_4_ [65] and proteins turned blue after staining with TB (Figure 4a,b,d–f). Typical idioblast cells were localized only in the adaxial epidermises of leaves in both environments (Figure 4a,c,d). Initially, idioblast cells look like large cells whose vacuoles accumulate phenolic compounds. Then, in these cells, between the cell wall and the plasmalemma, the synthesis of mucilage begins, which is mainly formed by proteins and, possibly, also by polysaccharides. At some phase of development, the idioblast looks like a cell just after transverse division, when the mucilaginous content of the idioblast presses the phenol-containing vacuole to the cell wall (Figure 4d, lower tab). Typically, when the volume of mucilage increases, the volume of the cytoplasm and phenolic vacuole either decreases or disappears (Figure 4d, upper tab). In the leaves and flowers of many dicots, epidermal mucilaginous idioblasts, mainly composed of pectins and sometimes proteins that accumulate between the primary cell wall and plasmalemma, have been described [66,67]. However, there is only one report concerning similar mucilaginous idioblasts in *Polygonum* species of the *Polygonaceae* family [68].

In the leaves of *F. tataricum,* besides the adaxial epidermis, phenolics were also detected in the guard cells of the stomata (Figure 4a), the vacuoles of non-glandular unicellular trichomes (Figure 4b), and multicellular glandular peltate trichomes (Figure 4e–h). Peltate trichomes were observed on both sides of the *in vivo* and *in vitro* leaves (Figure 4e,f), with the highest density on the abaxial side of the *in vivo* leaves. It is important to note that the accumulation of phenolics in the cells of the vascular bundle sheath was observed only in the *in vitro* leaves (Figure 4a,b,d). Other significant anatomical differences were the “openness” of stomata, whose pores were larger in the *in vitro* leaves, especially on the lower side of the leaf (Figure 4a,b), and the number of idioblasts with CaOx druses, which were rare on histological sections of *in vitro* leaves, but more abundant on sections of *in vivo* raised leaves (Figure 4a,f). 

On vital epidermal strips, the common morphology of epidermal cells of *in vitro* leaves was similar to those of *in vivo* leaves: the cells of the upper epidermis were oblong with slightly curved cell walls, whereas the cells of the lower epidermis had twisty cell walls and were connected to each other like pieces of a puzzle (Figure 5). The size of the stomata was increased under cultivation *in vitro* compared to outdoor conditions. If the leaves of *in vivo*-grown plants had stomata elliptical in shape, then the stomata of *in vitro*-cultured leaves were more rounded and larger, especially the stomata of the abaxial side (Figure 5). Their guard cells had a crescent shape and they were much more open in contrast to the semi-open stomata of the leaves of outdoor plants. 

The results of the epidermal and stomatal morphometry are summarized in Table 3. In *in vitro* leaves, the area of adaxial and abaxial epidermal cells was twice as large as that of epidermal cells *in vivo* (Table 3). In *in vivo* plants, ECA values were the same on both leaf sides, while, for the *in vitro* leaves, the area of adaxial epidermal cells was larger by 20% than the abaxial ones. The ECA was a highly plastic trait and characterized by significant PPI values (Table 3). In both environments, ECD was higher on the abaxial surface compared to the adaxial one, but, *in vivo*, the difference between both sides was 26%, while *in vitro* it was not statistically significant (Table 3). Furthermore, ECD values for *in vitro* leaves were much lower than those for *in vivo* leaves, by 51% and 84%, respectively, for the adaxial and abaxial sides. In both environments, the SD was higher on the abaxial side of the leaves. However, in *in vitro* conditions, SD decreased on both leaf sides compared to *in vivo* conditions, and it was stronger on the abaxial side of the leaf. Thus, on the abaxial side, the decline in ECD was higher and the decline in SD was lower than on the adaxial side, and this difference caused the SI value to rise only on the abaxial side of the leaf (Table 3). As a result, under *in vitro* conditions, the area occupied by stomata became greater just on the abaxial leaf surface. Thus, in the leaves of the *in vitro*-grown plants, the values of ECA, stomatal size, and SI were greater, while SD and ECD values were lower compared to *in vivo* conditions (Table 3). 

These results suggest that in *in vitro* leaves, the great expansion of epidermal cells (mainly pavement cells but also stomatal ones) was the factor causing the decrease in epidermal and stomatal densities and, in turn, a main contributory factor to the variability in the stomatal index. 

It is noteworthy that both sides of *in vitro* leaves had larger stomata than those of the *in vivo* leaves. Also, in the *in vivo* leaves, there were no differences in SL and SW between both sides of the leaf, whereas in the *in vitro* leaves, the stomata on the abaxial side had a length and width larger than those on the adaxial side. The length and width of the stomatal pores were also larger in *in vitro* leaves compared to those *in vivo*. In *in vivo*-grown leaves, the SPL of the adaxial epidermis was larger than that of the abaxial epidermis, but for *in vitro*-cultured leaves, there were no differences in SPL between both sides of the leaf. Stomatal pore width was slightly larger on the lower side of *in vivo* leaves, whereas for *in vitro* leaves, the SPW was much larger on the lower side compared to the upper side of the leaf. In the *in vivo* leaves, SPA did not differ significantly between lower and upper sides (31.89 and 29.59 mm2, respectively). between both leaf sides. In contrast, in the *in vitro* leaves, SPA on the abaxial epidermis was much larger than on the adaxial epidermis (59.98 and 82.41 mm^2^, respectively). Hence, the SPA of leaves *in vitro* was 1.85–2.79 times greater (for the upper and lower sides, respectively) compared to leaves *in vivo* and was the trait that underwent the greatest change and had the highest PPI.

Principal component analysis (PCA) was applied to nine epidermal and stomatal characteristics of the buckwheat leaves grown *in vivo* and *in vitro*: ECA, ECD, SD, SI, SW, SL, SPL, SPW, and SPA. It was found that Principal Component 1 (PC 1) explained 48.1% of the differences between the data and was determined by most of the stomatal characteristics (SL, SW, SPL, SPW, SPA) as well as ECA and ECD, but it had a strong correlation (>|0.4|) only with SPA (0.44), SPW (0.42), and ECD (−0.40) (Appendix A). 

As shown in Figure 6, the PC1 related to growth conditions because the data were distributed on the positive side (*in vitro* plants) and the negative side (*in vivo* plants) according to abscissa. At the same time, the traits determined by Principal Component 2 (PC2) were distributed according to the adaxial–abaxial principle. PC2 explained another 26.8% of the differences between the objects and was mainly related to SD (0.63) and SI (0.59) (Appendix A). 

Thus, the main traits of epidermal and stomatal characteristics that explained the difference between *in vivo* and *in vitro* plants were SPW, SPA, and ECD. The differences explained by belonging to the adaxial and abaxial sides were mainly determined by SD and SI.

Thus, morphologically, stomata of leaves cultured *in vitro* could be characterized as “open” (for the adaxial side) and even “fully open” (for the abaxial side), whereas stomata developing in *in vivo* conditions could be considered only “semi-open”. Changes in stomatal morphology in *in vitro*-propagated plants have been reported in several species, such as apples [69], *Delphinium* [70], roses [71], and *Citrus* [72], and these crescent-type stomata were associated with the inability to close in response to darkness and ABA treatment [70,73].

We also found that micropropagated plants of Tartary buckwheat that were placed in darkness showed stomata closure only on the 5th day of the culture in the dark (Figure 7). Spraying leaves with 50 µM ABA did not produce unambiguous results as the data of several experiments had significant variability.

### 3.3. Transmission Electron Microscopy (TEM) Examination of Epidermal Cell Walls in the Leaves of F. tataricum Grown In Vivo and In Vitro

The thickness of the cuticles and epicuticular waxes was shown to decrease in leaves of the plants cultured *in vitro* [37,74]. On histological sections of the *in vivo* leaves of Tartary buckwheat, the outermost cell walls of the adaxial epidermis had a thicker cuticle than those of the abaxial epidermis (Figure 4c,e); however, for epidermal cells *in vitro* at the level of histological sections, such a difference was not found (Figure 4d,f). To make a reliable measurement of the thickness of the cell wall and cuticle and to study their fine structure, we used TEM.

Electron microphotographs showed that under *in vivo* conditions, the adaxial epidermis of the leaves of Tartary buckwheat had a ridge-forming cuticle, and its outermost cell walls were thicker than those of the abaxial one (Figure 8a–d; Table 4). The difference in the thickness of the cell walls between adaxial and abaxial leaf surfaces was not so prominent (20%) (Table 4). In *in vitro* conditions, the outermost cell walls of the adaxial epidermis had the same thickness as those *in vivo*, but their cuticular ridges were more smooth and rare (Figure 8a,c,e,g; Table 4), while the cell walls of the abaxial epidermis were significantly thinner (almost 1.5 times) than those *in vivo* (Figure 8e,f; Table 4). The differences in cell wall thickness between both sides of the *in vitro* leaf were 43%. 

In *in vivo* conditions, the cuticle of the adaxial epidermis was almost twice as thick compared to that of the abaxial epidermis (Figure 8a–d; Table 4), but, under *in vitro* conditions, the cuticle of the adaxial epidermis was only a little thicker than that of the abaxial epidermis (Figure 8e–h; Table 4). Under *in vitro* conditions, the thickness of the cuticle of the adaxial epidermis decreased, whereas the thickness of the cuticle of the abaxial epidermis increased compared to that in the *in vivo* environment (Table 4). Moreover, in the *in vivo* leaves, the cuticle looked like a dense film, whereas in the leaves of micropropagated plants, it looked more like a loose layer. Importantly, the discontinuities in the cuticles and disturbances in cell wall structures, as well as the caverns between the cuticles and the cell walls, were observed only in the epidermal cells of the *in vitro* leaves (Figure 8f–h). 

Epicuticular waxes formed a thin network on both sides of the leaves of the outdoor plants (Figure 8c,d). In leaves of micropropagated plants, the surface of epidermal cells was covered by clumps of dense material or an amorphous layer, possibly formed by waxes and polysaccharides (Figure 8g,h). As shown by TEM images, the epicuticular layer in the leaves of micropropagated plants was not thinner but rather more conspicuous and amorphous than in the leaves of outdoor plants. Its chemical composition may have been more complex and additionally included cutin, polysaccharides, and phenolics; the latter are abundant in epidermal cell vacuoles.

### 3.4. Toluidine Blue Assay of Cuticle Permeability of Epidermal Cells in Leaves of Plants Grown in Different Conditions

To assay the possible defects in the cuticles and compare their permeability between *in vivo*- and *in vitro*-grown leaves, the TB test was used. Only little, rounded structures were found on the surface of the *in vivo* leaves using a dye: a few on the adaxial side (Figure 9a) and numerous on the abaxial side (Figure 9b). A study of the epidermal strips showed that TB penetrated only peltate trichomes (Figure 9c,d). The optimal time of submersion of the leaf in the dye solution was 20–30 min; in the case of exposure of less than 10 min, the trichomes were not stained. Longer staining (for 1 h) did not result in the coloring of stomata or pavement cells. Thus, in the leaves of plants *in vivo*, the cuticles of peltate trichomes were the only ones permeable to the TB solution.

The pattern of TB staining on *in vitro*-grown leaves was distinct, with the production of purple “spots” that were detected on *in vivo*-grown leaves (Figure 9e,f,j,k,m–p). The intensity of TB staining and the size and density of spots depended on the time of leaf submersion in the dye solution and the side of the leaf. The appearance of spots occurred after 5–7 min of TB staining, and the number of spots on the abaxial surface was much greater than on the adaxial surface, on which there were only a few spots (Figure 9e,f). Using epidermal strips, we revealed that the “spots” were formed by several nearby located stomata, often with different color intensities between them and the surrounding pavement cells (Figure 9g,i). At the adaxial surface, the majority of stomata and pavement cells were uncolored (Figure 9h). Almost all peltate trichomes were not stained by TB after 7 min of submersion in the dye (Figure 9i). With extending the staining of leaves in the dye solution to 10 min, an increase in the size and density of the spots on the abaxial leaf side was observed (Figure 9j–l). After 30 min of TB staining, the adaxial surface of the leaf remained generally uncolored (Figure 9m,n), except for single small, rounded structures (Figure 9n), while the abaxial leaf surface was almost completely colored by numerous dark spots (Figure 9o,p). On the adaxial epidermis, after 30 min of staining in the TB solution, we detected the coloring of only peltate trichomes (Figure 9q). On preparations of abaxial epidermal strips, even after 30 min of staining, several stomata and pavement cells were still uncolored or lightly colored (Figure 9r,t). However, the majority of guard cells of the stomata and all peltate trichomes, as well as cell walls and small structures inside the pavement cells, were stained (Figure 9r,s). This may indicate heterogeneity of the cuticle structures in the rows of stomata and pavement cells. 

Based on the results of the TB test (Figure 9) and the data from the TEM (Table 4, Figure 8), we can conclude the following: The abaxial epidermis of leaves *in vitro* had a defective cuticle, the permeability of which was significantly higher than the permeability of the adaxial cuticle or the cuticle of the *in vivo* leaf.Zones (“spots”) with a permeable cuticle were numerous on the abaxial side of the leaf and localized within groups of stomata surrounded by pavement cells. No such “spots” were found on the leaves of plants grown *in vivo*.The permeability of the cuticle of peltate trichomes, based on the time of staining, had no significant differences between the leaves either *in vitro* or *in vivo*.The thickness of the cuticle did not correlate with its permeability to dye since, according to TEM data, the thickness of the cuticle on the abaxial side of the *in vivo* leaves was less than that on the abaxial side of the *in vitro* leaves (Table 4, Figure 8b,d,f,h). Thus, the permeability of the cuticle was primarily determined not by its thickness but, most likely, by its composition and structure. According to TEM, in the leaves of micropropagated plants, the cuticle looked more like a loose layer.In *in vitro* leaves, the cuticle of the adaxial epidermis was less penetrable for TB than the cuticle of the abaxial epidermis. The permeability of abaxial epidermal cells to TB was most likely induced not only by faults in the cuticle structure but also by disturbances in the cell wall structure (Figure 8f) and the caverns between the cuticles and the cell walls (Figure 8h).In the leaves of outdoor plants, there were no differences in the TB staining between both leaf sides (only peltate trichomes were stained).

### 3.5. Calcium Oxalate Druse Examination in the Leaves of F. tataricum Grown In Vivo and In Vitro

Another significant difference between Tartary buckwheat leaves grown *in vivo* and *in vitro* was the frequency of occurrence and the size of CaOx druses (spherical conglomerates of individual crystals or “sphæraphides”). Such cells are very common in the genus *Polygonaceae* and were found in 120 of the 153 species investigated [43,68]. On histological sections of leaves *in vivo*, idioblast cells with CaOx druses were clearly visible, each with one crystal (Figure 10). *Calcium oxalate* druses were much more abundant on histological sections of *in vivo* leaves than on those of *in vitro* leaves and were localized within the spongy mesophyll near the abaxial epidermis (Figure 10a,b,e,f). Using hand-made vein preparations with an adjacent epidermis and spongy mesophyll, we found that the druses looked like complex crystals that occurred in enlarged cells confined to the mesophyll cell layer lateral to the vascular network (Figure 10a,b). *Calcium oxalate* idioblasts were either achlorophyllous cells (Figure 10c) or cells with a few plastids, which was clearly noticeable after the druses were dissolved by the addition of an EDTA solution (Figure 10e,f). In the *in vivo* leaves, CaOx druses were large and numerous (Figure 10a) and were found in every leaf examined. In leaves cultured *in vitro*, druses were much smaller, sporadic, and found only in 25% of the leaves investigated (Figure 10b, Table 5). The paraveinal layer was the most conspicuous and major zone of CaOx druse deposition, but additional, small—circa 2 µm—cuboidal crystals (‘prismatics’) were also detected in unicellular trichomes (Figure 10g,h; Appendix A). 

The density of CaOx druse distribution along the veins and the average area of the druses were three times lower and 2.1-fold smaller, respectively, in the *in vitro* leaves than in the *in vivo* leaves (Table 5). PPI values for the density of the druses (0.65) and for the area of the druses (0.55) were significant (Table 5).

To analyze the distribution of CaOx druses depending on their area, we ranged them into 15 groups from 0 to 1500 μm^2^ (the smallest druse area was 4.63 μm^2^ and the largest was 1485 μm^2^). Then, the data were normalized, since the mathematical sampling for data *in vitro* was smaller than for data *in vivo* (Figure 11). As shown in Figure 11, the seven groups of CaOx druse area classes (from 0–100 µm^2^ to 601–700 µm^2^) were common for both variants of leaves; other groups with a higher range of CaOx druse area were specific only for *in vivo* leaves (Figure 11). Additionally, *in vitro* and *in vivo* leaves differed in the main class of druses in accordance to their area: for *in vitro* leaves, the main class ranged from 101 to 200 µm^2^, while for *in vivo* leaves, it ranged from 301 to 400 µm^2^ (Figure 11). Druses with an area of 701–1500 µm^2^ were detected only in the leaves of outdoor plants. 

Later, we examined whether the number and sizes of druses in leaves of *in vitro*-grown plants were dependent on the concentration of CaCl_2_ in the medium for plant micropropagation, and grew the plants for 30 and 45 days on an MS medium with double CaCl_2_ concentrations compared to the standard MS medium. It was found that cultivation on a 2-fold CaCl_2_ medium did not significantly affect the sizes of the leaves, although there was a proclivity to extend their area and width. The sizes and density of druses were shown to be dependent on the Ca concentration in the nutrient medium because we revealed a rise both in the CaOx druses density (from 75.46 ± 5.18 to 95.47 ± 8.44) and in the CaOx druses area in leaves of plants grown on a double calcium medium (Figure 11). 

Surprisingly, when the duration of plant culture on a double CaCl_2_ medium was increased to 45 days, druses with an area of 600–1200 µm^2^, which were absent in the leaves of plants cultured on the same medium but for 30 days, were found (Figure 11). Moreover, the main class of druses became 201 to 300 µm^2^ (Figure 11). When the plants were cultured for 45 days on a control MS medium, the increase in druse area was not revealed; in contrast, druses larger than 500 µm^2^ were absent and the main class of druse area remained the same (101 to 200 µm^2^) as for the leaves of plants cultured on a standard MS medium for 30 days. Importantly, the plants cultured for 45 days on the medium with a standard CaCl_2_ concentration showed signs of senescence, such as shoot tip necrosis (tip burn), which might be the appearance of Ca deficiency [75]. This disorder was not observed in plants cultured for 45 days on a double CaCl_2_ medium. 

Hence, we can conclude that the number and area of CaOx druses in the leaves of cultured plants of Tartary buckwheat depended on both the calcium concentration in the medium and the duration of the culture period. It should be noted that even the culture of plants on a double CaCl_2_ medium for 45 days did not shift the area of druses in their leaves drastically and it should be noted that even the culture of plants on a double CaCl_2_ medium for 45 days did not shift the area of druses in their leaves drastically (only from 101–200 μm^2^ to 201–300 μm^2^) and did not reach the main class of druses area in leaves of outdoor plants (301–400 μm^2^). (Figure 11).

### 3.6. Estimation of the Phenotypic Plasticity of Fagopyrum tataricum Leaves in Response to In Vitro Conditions

An index of phenotypic plasticity ranging from zero to one was calculated for each of the twenty investigated traits of leaves; for some epidermal and stomatal traits, it was estimated for the adaxial and abaxial sides separately (Appendix A).

The uppermost PPI values (0.43–0.93) were found for leaf size, leaf area, palisade parenchyma size, ECA, ECD, SPWab, SPA, CaOx druse area, and CaOx druse density. Other variables such as leaf dry mass, SDad, SIab, SWab, cell wall thickness (ab), and cuticle thickness (ab) had less significant PPI values (0.23–0.32). The smallest plasticity indices were for leaf thickness, SDab, SLab, and SPLab (0.1–0.16). For most epidermal and stomatal characteristics for the abaxial side of the leaf, the PPI was more prominent than for the adaxial side: they were ECD, SI, SL, SW, SPL, SPW, SPA, cell wall thickness, and cuticle thickness. In a few cases, the appearance of the plasticity of the trait was detected only for the abaxial side of the leaf (SL, SPL, cell wall thickness). In contrast, for ECA and SD, the PPI was greater for the adaxial side compared to the abaxial one. It was intriguing to find that the PPI values for cuticle thickness varied more for the abaxial side than for the adaxial side and, at the same time, the trait itself was expressed in the opposite direction on both sides of the leaf—increasing on the abaxial side and decreasing on the adaxial side (Table 4).

### 3.7. Apoplast Washing Fluid Isolation from High-Humidity-Treated Leaves of Outdoor Plants

Since the main factor that appeared to affect the success of the VIC procedure for *in vitro* leaves was the openness of stomata, we attempted to establish high humidity conditions using a “wet chamber” for outdoor plants to “open” their stomata and improve the VIC technique for them (Appendix A). After 2.5 h of treatment, the increased opening of stomata in the experimental leaves compared to control ones was observed. As expected, for untreated leaves, AWF of a yellowish color and a precipitate after centrifugation of AWF at 10,000× *g* were found. Using the “wet chamber”, no change in AWF color or the formation of the precipitate was shown (Figure 12). For outdoor leaves, similar data were received when 600× *g* and 1500× *g* forces of centrifugation were used as well. 

## 4. Discussion

### 4.1. The Peculiarities of the VIC Procedure for Tartary Buckwheat Leaves

The VIC method suggested for buckwheat leaves consists of a few consecutive stages (Figure 1), each of which can affect AWF purity. Leaf collection and preparation are the first important stages. It should take as little time as possible. Outdoor plants should be watered in advance. This will help prevent the loss of leaf turgor and stomatal closure. In some cases, it is recommended that a detergent treatment be applied before the leaf infiltration step [76]. In general, the need for detergent treatment depends on such morphological features as the openness of stomata, the density of trichomes, and the thickness of the epicuticular layer [29]. Leaves *in vivo* have a thin and dense epicuticular layer (Figure 4), so leaf rinsing with a 0.1% Tween solution reduces the surface tension of the leaf blade and ultimately facilitates the infiltration procedure, helping to shorten the infiltration stage. For the leaves of *F. tataricum* grown *in vitro*, such a procedure was unnecessary because they had a rather friable epicuticular layer on both sides of the leaves and open stomata. As a rule, there is an inverse correlation between stomatal conductance and the number of infiltration steps necessary to reach the constant weight of the infiltrated leaf [10]. This seems to be the main reason for the longer infiltration procedure for plants *in vivo* since their stomata were semi-open, whereas leaves *in vitro* had open stomata. 

The effectiveness of the VIC method seems to be highly dependent on the appropriate device for AWF collection. The principle of such a device is the same: a collector for gathering AWF is placed at the bottom of the centrifuge tube, while the leaves rolled with Parafilm or Miracloth are situated above the collector, in the upper part of the tube [11,14,77]. Rolling the leaves with a protective material prevents them from being damaged [11]. For rice leaves, in order to minimize leaf sticking, each leaf segment of a length of about 5 cm is inserted into a pipette tip of 1 mL and only then put into a centrifuge tube [31]. 

A soft plastic grid used in VIC devices for buckwheat, unlike Parafilm and Miracloth, does not prevent either infiltration or AWF outflow and keeps the leaves from deforming. Moreover, the leaves can be easily taken out of the syringe after the infiltration and transferred into the AWF collection apparatus. For small leaves of aseptically grown plants, which are very sensitive to dryness, this is the only acceptable way to minimize damage. The next important nuance is that the centrifugation of infiltrated leaves should be performed with a central axis in the form of a 2 mL plastic syringe. This allows the leaves to maintain a stable position in the grid, prevents them from being damaged, and does not clog the hole of the AWF collector. To make the central axis, automatic pipette tips can also be used in VIC devices [14]. 

The optimal centrifugation speed is altered for different objects and organs, and it is also varied for plants of different ages and physiological states. Nouchi et al. [31] showed that rice leaves have a highly hydrophobic surface and stomata with a small aperture, and these characteristics make it impossible to collect AWF at a lower centrifugation rate than 6000× *g*. Nevertheless, the use of higher than 600–900× *g* centrifugation rates remains debatable. It is assumed that to get AWF without cytoplasmic contamination, the rate of centrifugation should not exceed 1000× *g* [10,11].

Any investigation using AWF needs a reliable method to detect cytoplasmic contamination. Ideally, it should be simple and fast, preferably cheap, with a minimum volume of the sample. It can even be conducted visually, that is, by the express method. The most commonly used method for determining the degree of contamination of AWF is to assess the presence/absence of cytoplasmic enzymes such as MDH, hexose phosphatisomerase, and glucose-6-phosphate dehydrogenase [10,11,14]. It is considered acceptable if the MDH activity in the AWF does not exceed 3% of the intracellular MDH activity [10,78]. However, the measurement of MDH activity is not sensitive enough to be a reliable indicator of AWF purity, and it is also time- and material-consuming. Some authors use quick visual control of AWF. For example, Andreasson et al. [76] pointed out that AWF from healthy leaves should be almost colorless; Dani et al. [78] and Marentes et al. [79] simply discarded any green-colored AWF. Obviously, the use of visual control of chlorophyll contamination of AWF as a damage indicator is more quick and sensitive than MDH enzymatic control; it can be used as an express method to fine-tune the VIC procedure.

In the stage of AWF isolation, infiltrated leaves of *in vivo* buckwheat plants required gentler centrifugation than leaves *in vitro*. This was unexpected since the leaves of the outdoor plants were thicker and looked more vigorous than the small leaves *in vitro*. Nevertheless, the cell damage, as judged by the green plaque on the walls of tubes after AWF centrifugation; the appearance of mesophyll cells and trichomes in the sediment; and an increase in protein yield and MDH activity were detected only for leaves *in vivo* at 900× *g* (in some cases) and as a regularity at centrifugation of 1500–3000× *g*. The main reason for such disorders was the openness of stomata, which had great distinctions between plants grown *in vitro* and outdoors. The SPA of leaves *in vivo* was much smaller compared to leaves *in vitro* (Table 3). The dependence of filtration and centrifugation efficiency on the openness of stomata has been observed in rice leaves [31]. Another reason for cell damage may be that large CaOx druses were found in abundance along leaf veins and among spongy mesophyll cells in leaves *in vivo* (Figure 4a,e and Figure 10a,c,e). Because of the semi-closed stomata and high centrifugation forces of 900–3000 g, the exerted pressure of the infiltrated buffer might have destroyed not only the stomata guard cells but also the mesophyll cells and idioblasts; heavy starch grains and sharp druses released from them were additional damaging factors. In contrast to the leaves of outdoor plants, *in vitro*-grown leaves had open stomata and contained a minor number of idioblasts with CaOx druses of a smaller size compared to leaves *in vivo* (Figure 10b,d). Thus, under the same centrifugation conditions, it was possible to get more AWF and more protein from *in vitro* leaves than from *in vivo* leaves. Moreover, a higher centrifugation force (1500× *g*) could be applied to *in vitro* leaves without the appearance of yellowish AWF and a green precipitate during the VIC procedure. For leaves *in vivo*, this result could be obtained mainly at a 600× *g* centrifugation force.

As outlined above, the size of stomata pores was the determining factor in obtaining AWF free of chlorophyll contamination. Considering that the primary factor influencing stomatal opening *in vitro* was high RH, we attempted to create a comparable environment for outdoor plants by keeping them in a “wet chamber” for 2–2.5 h. This time was enough to “open” the stomata and get a colorless AWF after the centrifugation of infiltrated leaves at 900× *g* (Figure 12). Thus, this relatively simple approach might significantly improve the quality of AWF collected by the VIC method. 

In summary, several conditions should be fulfilled for the successful isolation of AWF from buckwheat leaves by the VIC method. For *in vivo* plants, the physiological state of the plant is very important, especially the leaf turgor and the openness of the stomata. Outdoor buckwheat leaves have closed or semi-open stomata, so we strongly recommend checking the state of stomatal openness before starting the VIC procedure. To improve stomatal opening, the soil or substrate in the pots should be abundantly watered 1 h before VIC. If the stomata remain closed after watering, a “wet chamber” can be additionally used (you need to cover watered plants with a transparent container for 2–2.5 h). On a sunny day, it is better to move the plants into the shade or cover them from direct light by the canopy before the VIC. It is not recommended to perform VIC with leaves of outdoor plants if the air temperature is above 28 °C since thin buckwheat leaves lose turgor and become withering, which is a protective reaction to restrict transpiration. For the isolation of AWF from *in vivo* leaves, it is optimal to use 2–3 leaves 3–4 cm wide of approximately 1-month-old plants (4–5 weeks, depending on weather conditions, until inflorescence formation). For *in vitro* plants, use leaves of 1-month-old plants (28–32 days old), 1 cm or more wide, with a well-opened leaf plate.

After cutting from plants, rinse leaves with water to remove surface contaminants and exudates; *in vivo* leaves should be additionally rinsed with a 0.1% Tween 20 solution. In the VIC method applied to buckwheat leaves, manual leaf infiltration with a 20 mL syringe was used, which is gentler than using a vacuum pump. Leaves should be infiltrated with a neutral or slightly acidic buffer (100 mM Na-phosphate buffer, pH 6.5) supplemented with protease inhibitors (10 μM PMSF and 5 mM EDTA). At the stages of infiltration and centrifugation, it is necessary to use a soft plastic grid that does not prevent AWF flow and protects leaves from deformation and damage. *In vitro* leaves require a plastic grid to be used immediately after cutting from the plant, before the rinsing. To collect AWF, use a system consisting of several plastic tubes and syringes of different sizes assembled as a “nesting cup”, as shown in Figure 1. The most optimal centrifugation mode for the isolation of AWF from *in vivo* leaves is 600× *g*, 10 min and 900× *g*, 10 min for *in vitro* leaves. After centrifugation of the AWF at 10,000× *g*, you need to make a visual control of a green precipitate on the walls and bottom of the Eppendorf tube to quickly check the AWF purity. Eppendorf tubes with strong green precipitate on the walls and bottom should be discarded. The resulting volume of AWF should be lyophilized and stored at −20 °C or −70 °C before the start of proteomic analysis.

### 4.2. The Anatomical Structure of Tartary Buckwheat Leaves

The plants use their leaves as sensors and gates to communicate with the surrounding environment. They allow plants to absorb sunlight during photosynthesis, exchange CO_2_ and O_2_ with the environment, and control temperature and vapor pressure deficit (VPD) by removing water vapor through stomata during transpiration. The shape, size, and anatomy of leaves are closely controlled by complex gene regulatory networks during development and in response to a variety of environmental factors, such as the quality and intensity of lighting, temperature, or availability of water [80]. Broad phenotypic plasticity, based on the action of these multifaceted, partially overlapping modules, is crucial for plant fitness because plants cannot avoid drastically changed or unfavorable environments, and the only way for them to survive and adapt to such environments is to rebuild their bodies and/or metabolism.

Although Tartary buckwheat is a plant rich in secondary compounds, and its metabolomics and genetics have been extensively investigated [45,46,47,48,49,50], the anatomical features of the buckwheat leaf have only been studied in one article [43], and we have clarified them. Tartary buckwheat leaves are both dorsiventral, i.e., bifacial when upper and lower leaf sides are different in mesophyll structure, and amphistomatic, with anomocytic stomata scattered throughout both leaf surfaces. In amphistomatic, dorsiventral leaves of angiosperms, the adaxial and abaxial parts have different anatomical complexity due to certain locations of pallisade and spongy mesophyll, distinct densities of bilaterally arranged stomata [81], and, often, different cuticular thicknesses [82]. Also, both leaf surfaces may be contrasting in the number of morphologically and biochemically distinct trichomes and idioblasts that protect the leaves from diverse unfavorable environmental factors [83]. The upper surface of the Tartary buckwheat leaf had a larger ECA and longer stomata but lower ECD, SD, and SI compared to the abaxial surface (Table 3). In leaves of Tartary buckwheat, not only the cuticle but also the outer cell walls of the adaxial epidermis were thicker than those of the abaxial epidermis (Figure 8; Table 4). Additionally, the cuticle of the adaxial epidermis produced folds (“ridges” or “wrinkles”) and was two times thicker than that of the abaxial epidermis. In tea leaves, the differences in cuticle thickness between adaxial and abaxial epidermises correlated with a stronger cuticular transpiration barrier [82]. To be more accurate, cuticle conductivity appeared to be regulated not by the cuticle thickness but rather by its composition and structure [82,84], as well as the presence of trichomes and stomata, around which were found the largest number of aqueous pores [85].

Tartary buckwheat leaves have glandular peltate trichomes on their surface, which accumulate phenolic compounds and are arranged bilaterally. The density of trichomes is higher on the abaxial side and lower on the adaxial side (Figure 9a–d), which appears to correlate with stomatal density. Previously, peltate trichomes were found just on the abaxial surface of Tartary buckwheat leaves [43].

Epidermal cells on both leaf sides had different morphology and different abilities to synthesize anthocyanins (Figure 1a). It was argued that anthocyanins, mostly localized in epidermal and subepidermal tissues of adaxial leaf surfaces, act as UV and highlight screens and also as antioxidants to protect underlying photosynthetic tissues from photoinhibition and oxidative stress [86,87,88]. 

Moreover, both sides of the Tartary leaf have their own idioblasts: mucilaginous-containing idioblasts of unclear function in the adaxial epidermis (Figure 4a,c,d) and CaOx-containing idioblasts located between spongy parenchyma cells and around vessels near the abaxial epidermis (Figure 4e and Figure 10a–f). In recent years, numerous studies have elucidated the physiological role of CaOx and its involvement in calcium homeostasis, “alarm” photosynthesis, metal detoxification, endosymbiosis with beneficial bacteria, herbivore protection, and other processes (reviewed by [50,89,90]). 

### 4.3. Abaxial–Adaxial Polarity of Tartary Buckwheat Leaves

The results of this study showed that, *in vitro*, a broad set of morphological and anatomical traits of Tartary buckwheat leaves tend to change together, demonstrating a specific form of phenotypic plasticity in response to unusual growing conditions. It was proposed that adaptive plasticity in functional traits facilitates rapid adaptation to new conditions [36]. Among the key traits suggested for the investigation of adaptive phenotypic plasticity to climate change are leaf size, shape, and thickness; leaf mass per unit area; stomatal size and density; and leaf pigmentation [36]. Most of them were changed in the leaves of Tartary buckwheat plants in cultures *in vitro* (Appendix A).

It is notable that in Tartary buckwheat, the development of some leaf traits under *in vitro* conditions was suppressed (leaf length, width, area, thickness, and dry mass; palisade mesophyll length and width; ECD and SD (on both leaf sides); cell wall thickness (on the abaxial side) and cuticle thickness (on the adaxial side); CaOx druse area and density). Other features, on the other hand, exhibited higher values *in vitro* than *in vivo*: ECA, SL, SW, SPW, and SPA (on both leaf sides) and SPL, SI, and cuticle thickness (on the abaxial side).

The changes in the epidermal and stomatal characteristics of the adaxial and abaxial sides of the leaves had different expressions and directions, which were reflected in variable PPIs. In general, the abaxial side of the leaf underwent larger changes in response to *in vitro* conditions (Appendix A). Similarly, in *in vivo* conditions, the abaxial guard cells were usually more sensitive to environmental signals such as fluctuations in light intensity [91] or light quality [92,93], soil water status [94], RH [95,96], and CO_2_ concentration [97,98]. Wang et al. [99] demonstrated that abaxial stomata of *Vicia faba* were more sensitive to Ca^2+^ and ABA than adaxial stomata in their opening and closing. There are observations indicating that mechanisms of regulation for H^+^ pumps and K^+^ channels in the guard cells of opposite types of stomatal cells may be different [99,100]. The establishment of abaxial–adaxial polarity starts shortly after the initiation of leaf primordium, and this program is governed by the activation or inhibition of different components of leaf regulatory modules combining regulatory genes, transcription factors, proteins, and microRNAs [80]. Probably, specificity in the regulation of signal transduction pathways that control the movements of abaxial and adaxial stomata is a component of the unique program of regulation of adaxial–abaxial polarity formation and the establishment of physiological functionality on the two opposite sides of the leaves.

Field and outdoor plants are influenced by a complex of different abiotic and biotic factors. Many of these environmental factors do not affect plants *in vitro* or their action does not have the same quality, amplitude, or variability as *in vivo*. Plants grown *in vitro* are influenced by a set of factors that never act together on plants *in vivo*. These are high RH (≥85%), low light intensity, light spectrum different from the solar spectrum, the low level of CO_2_ in the vessels, heterotrophic nutrition, cut wounds under repeated micropropagation cycles, the special mineral composition of the nutrient medium, and others [37,38].

In this study, outdoor buckwheat plants and micropropagated plants were grown under extremely distinct environments, notably in terms of the intensity of irradiance and values of RH. Analyzing the alterations observed in different traits of Tartary buckwheat leaves under *in vitro* conditions, we aimed to highlight those that might be caused by low lighting and high RH, the common factors that have the strongest influence on the morphology, anatomy, physiology, and biochemistry of *in vitro*-cultured plants [37,38].

### 4.4. Phenotypic Plasticity Governed by Low Lighting

It was suggested that some structural and morphological modifications observed in the leaves of Tartary buckwheat plants cultivated *in vitro* could be provoked by typically “shaded” conditions at PPFDs of 37 μmol m^−2^s^−1^, which were in the range of commonly applied irradiance for plant clonal micropropagation (PPFDs = 20–50 μmol m^−2^s^−1^). In *in vivo*-grown plants, shade or low light acclimation behavior mostly under 50–100 μmol m^−2^s^−1^ of PPFD appeared in developing larger leaves with thinner leaf lamina and a smaller number of stomata compared to sunlit leaves [101,102,103,104]. Leaf thinning was shown to occur as a result of a decrease in the thickness of the mesophyll parenchyma tissue caused mainly by a reduction in cell rows of palisade mesophyll [101,105,106] because its formation strongly depends on the light intensity [81]. The mesophyll parenchyma of shaded leaves was characterized by large intercellular spaces [106]. Generally, the stomatal and pore sizes (length and width) were smaller and SPA [104], SD, and SI [102,103,107] were lower in shaded leaves than in sunlit leaves. In part, this difference could have been caused by increasing the epidermal cell size during the shading-stimulated expansion of leaves [107]. Such morphological and anatomical changes are likely to compensate for reduced photosynthesis per unit of leaf area [108]. The size of the palisade mesophyll of shaded leaves was noted to be changed in different ways: not to change [101], to become larger [105], or to become smaller [109] than that of sunlit leaves.

In the leaves of buckwheat plants cultivated *in vitro*, the number of rows of palisade mesophyll cells was reduced to one or two, compared to 3–4 rows in the leaves of outdoor plants. The decrease in the palisade mesophyll rows, which is often correlated with thinning of the leaf lamina, was the most characteristic change in the anatomy of the leaves of the *in vitro*-grown plants [110,111]. Surprisingly, the leaves of *in vitro*-cultured *Liquidambar styraciflua* [112] and strawberry [113] did not have any well-defined palisade layers.

It was shown that the increase in light intensity was correlated with mesophyll expansion and increases in leaf thickness, but oppositely correlated with epidermal cell expansion and leaf area [114]. In the leaves of micropropagated Tartary buckwheat plants, the area of epidermal cells enlarged nearly twice as much as the size of palisade mesophyll cells, which decreased by two times compared to leaves *in vivo.* Accordingly, with the expansion of epidermal cells, a decrease in epidermal and stomatal density occurred. However, these anatomical changes did not result in the enlargement of leaves and their size was only 6% of the area of the leaves of outdoor plants. The smaller size of the cells of the palisade mesophyll is typical for plants cultured *in vitro* compared to *in vivo*-grown plants [112,113,115]. The leaves of micropropagated Tartary buckwheat plants were also characterized by enlarged intercellular spaces, which is typical for shade-grown leaves, but their stomata were larger than the stomata of their *in vivo* counterparts and had widely open stomatal pores. These features principally distinguished the *in vitro* leaves from the *in vivo* leaves that had adapted to low-intensity light. 

The absence of anthocyanin pigmentation in the leaves of the buckwheat plants grown *in vitro* looked like an indicator of insufficient illumination and a shortcoming in UV, blue, and R-FR light under artificial lighting. Transcription factor HY5 is a downstream component of phytochrome, cryptochrome, and UV-B (UVR8) photoreceptor-mediated light signaling; these photoreceptors are activated by red, blue, and UV light, respectively [116,117]. HY5 was shown to be involved in anthocyanin biosynthesis through direct binding to the promoters of both early and late biosynthetic genes of flavonoid metabolism [116]. Thus, the absence of anthocyanin biosynthesis in the *in vitro* leaves could indicate the failure of HY5 activation by the above-mentioned photoreceptors. 

### 4.5. Phenotypic Plasticity Governed by High RH

The second most important abiotic factor that influences plant development *in vitro* is a high RH, which reaches 100% in culture vessels [118,119]. In natural or close-to-natural environments, high RH persists continually only in limited earthly places such as tropical forests, montane cloud forests, rainforests, and shady humid woodlands; the lack of good ventilation in greenhouses also contributes to the emergence of high RH [120]. The life of the plant is ensured by two primary processes: the absorption of CO_2_ by the leaves and the uptake of water by the roots. Carbon dioxide is absorbed through the stomatal pores of the leaves simultaneously with the transpiration of water vapor. The transpiration of water by stomata fuels the absorption of water by roots and its transport through the xylem. These two pumps, the upper one (water transpiration by the leaves) and the lower one (water absorption by the roots), always cooperate in water transport and control one another’s activity. Transpiration creates a water potential gradient from the roots to the leaves, and this gradient has a significant impact on the physiology of the plant [121,122]. In homoiohydric plants, the stomata, cuticle, intercellular spaces, and xylem are all parts of the machinery for maintaining plant hydration [123]. The main problem for plants cultivated in high moisture conditions is the decrease in transpiration levels [124,125,126,127], which is directly connected to the decrease in VPD under high RH. Low VPD slows down transport processes, including calcium transport, CO_2_ absorption, and plant metabolism, which, in turn, delays plant development. Regrettably, the data regarding the impact of high RH on leaf anatomy are scarce and insufficiently reliable as control and experimental plants, as a rule, are grown under low lighting, which corresponds to shading conditions [128]. Consequently, the leaves of control plants used in such experiments could exhibit features of adaptation to shade, being morphologically and anatomically different from leaves grown in sunlight or high-intensity light.

It is believed that at high RH, as with shading, plants tend to expand their leaf area and decrease leaf thickness, in the first case, to compensate for declining transpiration [125], and, in the second case, to improve photosynthesis [108]. Since transpiration and photosynthesis are both directly dependent on each other, it is not remarkable that the morphological responses of plants to elevated humidity and shade are similar.

In hygromorphic woodland herbs, high RH provoked the formation of larger leaves with extended epidermal cells as well as a lower stomatal density compared to herbs in low RH conditions. This indicates the stimulation of leaf expansion by high RH. In expanded leaves, the alteration in leaf anatomy was expressed in the enlargement of intercellular spaces in mesophyll tissue, but mesophyll cell sizes were not affected [129]. Thus, we can assume that a strong decrease in the size of the cells of the palisade mesophyll is a characteristic only of *in vitro* plants.

The most specific trait triggered by high RH was the formation of large, mainly increased in length, wide-open stomata that were less sensitive to stomata-closing stimuli [120]. The leaves of Tartary buckwheat plants cultivated *in vitro* were also characterized by widely open stomata that did not respond to darkness and reacted poorly to the treatment of ABA. It is suggested that stomatal dysfunction may be due to low endogenous ABA content in the leaves of plants grown at high RH [130], probably due to the inactivation of ABA (oxidation or conjugation) rather than the suppression of its biosynthesis [131,132,133]. Partially, it may be caused by disturbances in cuticles and cell walls, which are the components involved in the signaling systems of plant cells. Low endogenous ABA levels affect the composition of cuticles and cell walls in tomato ABA-deficient mutant *sitiens* [134]; cuticle permeability was shown to be enhanced in ABA-deficient mutants [134,135]. There exists a cross-talk between stomatal ABA signaling and signaling in cuticle and wax biosynthesis [136,137]. Wax-related genes, including CER6, have ABA-responsive elements in their promoters and are induced by ABA; thus, it is possible that ABA directly regulates wax synthesis (reviewed in [138]). Thus, ABA regulates a cuticular modification, while cuticular modification influences ABA signaling and biosynthesis, thereby impacting stomatal physiology and development.

Commonly, both shading (low lighting) and an increase in air humidity tend to decrease the amount of wax and the thickness of the cuticle [139]. A reduced cuticle thickness and reduced wax content were routinely observed in the leaves of the *in vitro* plants, normally grown under high RH and in low lighting [115,139]. In *Liquidambar styraciflua* leaves, cuticles were not developed at all [112].

The transport barrier of cuticles was shown to be sensitive to humidity [140]. The thickness and composition [139] as well as the mechanical properties [141] and permeability [142] of cuticles are affected by high RH environments. The permeances of isolated cuticles of *Prunus* for water flow were 2–3 times higher at 100% humidity than at 2% humidity [142]. In cuticles, hydrophilic cell wall materials such as cellulose, pectins, and hemicelluloses are thought to be the main sites of water absorption [143]. With a simple TB test used earlier to detect cuticle defects in cuticle mutants [144] and the relative cuticle permeability in ABA-deficient mutants [134], we demonstrated that in the *in vitro* leaves, the cuticles of the abaxial epidermises were much more penetrable for TB than the cuticles of the adaxial epidermises, as well as the cuticles of *in vivo* leaves. The permeability of abaxial epidermal cells to TB was most likely induced not only by faults in the cuticle structure but also by disturbances in the cell wall structure (Figure 8f) and the caverns between the cuticles and the cell walls that were detected under TEM study. In the *in vitro* leaves of Tartary buckwheat, zones of cells with a permeable cuticle detected as TB-colored spots were numerous on the abaxial side and formed by groups of stomata surrounded by pavement cells. No such “spots” were found on the leaves of plants grown *in vivo*. It appears that the transport barrier of the cuticles of the abaxial epidermises of the *in vitro* leaves was low, which suggests that the level of cuticular transpiration may be rather high. 

High RH was shown to directly (ABA-independently) affect leaf expansion [145] due to its effect on leaf water status, resulting in bigger stomatal and pavement cells [146]. The expansion of epidermal cells was much more pronounced than that of the guard cells of stomata probably because the turgor pressure in the guard cells was higher than in the pavement epidermal cells [147]. Since both sides of amphistomatic, isobilateral leaves have their own regulation and sufficient hydraulic isolation [148], this could provide the differences in stomatal sizes between abaxial and adaxial leaf surfaces. 

Thus, low lighting and high humidity cause similar changes in the morphology, anatomy, and physiology of plant leaves, both cultivated *in vitro* and *in vivo*. Their action on adaptive changes in the leaves of Tartary buckwheat cultured *in vitro* is shown in Figure 13. 

## 5. Conclusions and Future Directions

This study provides versatile insight into the morphological and physiological adaptation of plants to *in vitro* conditions governed mainly by two stressors: low lighting and high RH. In general, both of them act synergistically on the development of adaptive traits including thinning of the leaf blade, the enlargement of intercellular spaces, the expansion of epidermal cells, and cuticle and cell wall modifications (Figure 13). At first sight, an exception may be the opposite direction of the action of two main stressors on the size of stomata and SPA (Figure 13). However, under high RH, the key photoreceptor phyB, which is implicated in light stimulation of stomatal differentiation, is involved in ABA breakdown [149]. Thus, under high RH, phyB becomes a positive regulator of the formation of large, wide-open, and weakly responsive to closing stimuli stomata.

Despite extensive research on the effects of *in vitro* conditions on the physiology and biochemistry of micropropagated plants, certain important aspects of *in vitro* plant biology remain poorly investigated. In particular, there are no reliable data on the levels of transpiration of plants cultivated *in vitro*. It is reasonable to suppose that they are lower than those of plants cultivated *in vivo*, which can be indirectly judged by Ca transport and the density and size of CaOx druses in the idioblasts of *in vitro* buckwheat leaves. Even with a 2-fold increase in the CaCl_2_ content of the nutrient medium, the density and size of the druses did not reach the parameters of outdoor plants. In general, the results of the current study suggest that in Tartary buckwheat plants cultivated *in vitro*, structural and anatomical adaptations were mainly aimed at overcoming the negative action of low VPD by an overall increase in transpiration, with involvement in both stomatal transpiration and cuticular transpiration (Figure 13). As discussed earlier, under high RH or shading, plants compensate for the decrease in transpiration and photosynthesis by increasing the area of the leaf lamina. Because for *in vitro*-grown buckwheat plants, the area of leaf lamina is only 6% of that of outdoor plants, the partial compensation for the suppression of transpiration under low VPD appears to be achieved only by concurrently increasing stomatal and cuticular transpiration (Figure 13). Under *in vitro* conditions, an expansion of epidermal and stomatal cells was likely one of the important factors that increased both the area of stomatal pores and the cuticular permeability. In only a few studies has the enlargement of epidermal cells in leaves grown under high RH been either mentioned [127] or analyzed [146]. As stated above, for *in vitro*-grown plants, the morphometric studies of epidermal cells were not previously fulfilled. The use of TEM has revealed extensive rearrangements of outermost cell walls and cuticles in the epidermal cells of *in vitro* leaves, especially on the abaxial side. TB tests have shown that zones of cells with a permeable cuticle detected as TB-colored spots were numerous on the abaxial side of *in vitro* leaves and formed by groups of stomata and adjacent pavement cells. No such “spots” were found on the leaves of plants grown *in vivo*. 

Many of the listed changes in the leaves of plants cultivated *in vitro* (thinning of leaf lamina, enlargement of intercellular spaces, increase in stomata size and SPA, the reduced thickness and increased permeability of the cuticles of the epidermal cells, decrease in the density and sizes of CaOx druses) are favorable for the isolation of AWF with the use of the VIC method. They contribute to the greater effectiveness of VIC for *in vitro* leaves compared to *in vivo* ones. However, the main factor determining the yield of AWF and its purity was the openness of stomata. To increase the openness of stomata in outdoor plants, a simple approach using a wet chamber was suggested. To our knowledge, for *Fagopyrum* species, this is the first study to develop a VIC technique for the isolation of AWF from leaves. The elaborated technique lays the foundation for the secretome study of buckwheat. It is known that the cuticle and cell walls and the apoplast as a whole are a pool of regulatory proteins and peptides [150,151]. We can speculate that changes in the composition of the cuticle and cell walls may be involved in signaling, triggering anatomical, morphological, physiological, and biochemical changes in plants cultivated *in vitro*. In this case, plants cultivated *in vitro* could be a valuable object for foundational knowledge. 

## Figures and Tables

**Figure 1 plants-12-04048-f001:**
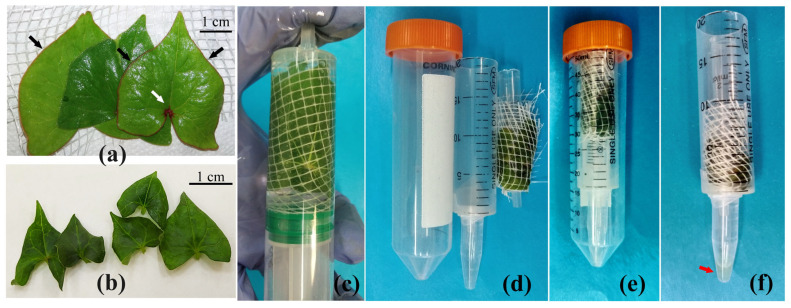
VIC procedure of AWF isolation from *in vitro* (**a**) and *in vivo* (**b**) Tartary buckwheat leaves. Infiltration step (**c**). The AWF collecting system in disassembled (**d**) and collected (**e**) states before centrifugation. The AWF at the bottom of the Eppendorf tube (red arrow) after centrifugation (**f**). The white arrow in (**a**) indicates the anthocyanin spot on the adaxial surface of the *in vivo* leaf, where veins were branching out from the petiole; black arrows indicate anthocyanin staining along the margin of the leaf blade (two of the three leaves showed this staining). In these places, anthocyanins were accumulated in the simple, non-secretory trichomes and epidermal cells. In vitro, leaves usually do not have these patterns of anthocyanin accumulation (**b**).

**Figure 2 plants-12-04048-f002:**
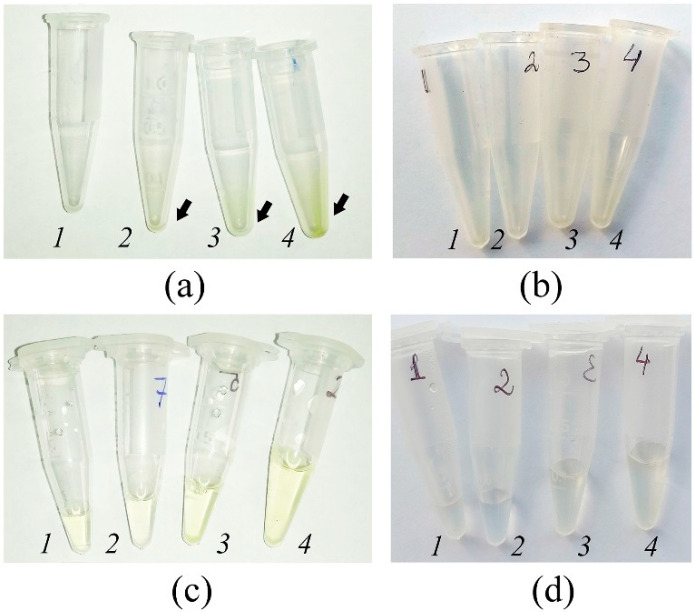
The effect of different centrifugation regimes (1—600× *g*, 2—900× *g*, 3—1500× *g*, and 4—3000× *g* for 10 min) on the appearance of green precipitates in AWF isolated from *in vivo* and *in vitro* leaves. The arrows in Figure 2a indicate the green precipitate appearance. All AWF samples were centrifuged finally at 10,000× *g*. (**a**,**b**) Empty Eppendorf tubes after 10,000× *g* centrifugation of AWF isolated from *in vivo* and *in vitro* leaves, respectively. The green precipitate on the tube walls is marked with an arrow. (**c**,**d**) The supernatants transferred to the new tubes after 10,000× *g* centrifugation of AWF isolated from *in vivo* and *in vitro* leaves, respectively.

**Figure 3 plants-12-04048-f003:**
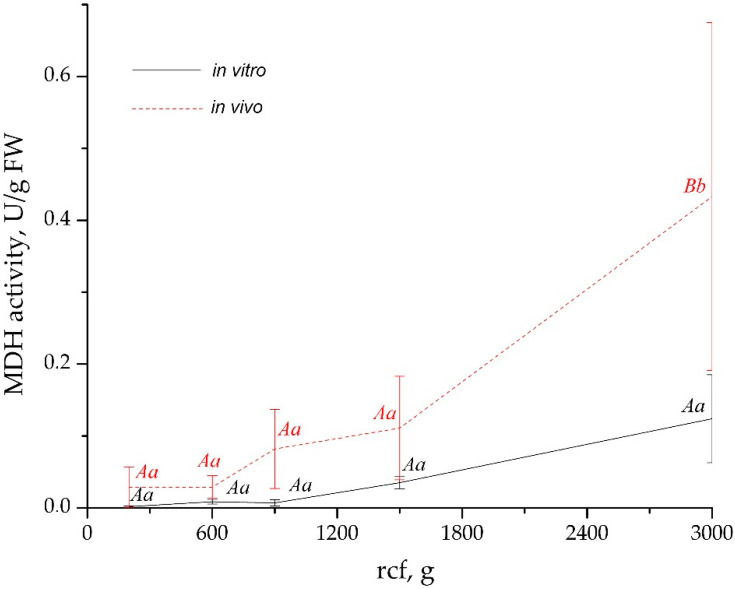
MDH activity in AWF samples gathered at different centrifugal forces from the *in vitro* and *in vivo* leaves. Significant differences between samples (*p* ≤ 0.05) are marked with different letters above the curve. Uppercase letters indicate the difference between the objects (leaves of *in vivo* and *in vitro* plants) at the same centrifugal force; lowercase letters indicate the effect of centrifugal force within the same object.

**Figure 4 plants-12-04048-f004:**
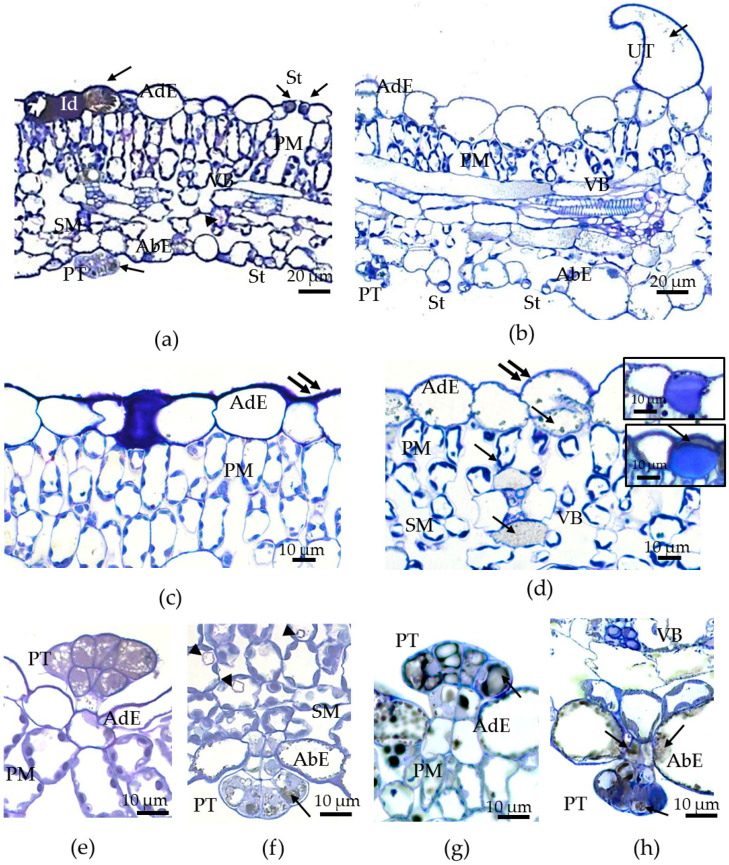
Histological characterization of Tartary buckwheat leaves grown *in vivo* and *in vitro*. (**a**) Cross-section of *in vivo* buckwheat leaf. (**b**) Cross-section of *in vitro* buckwheat leaf. (**c**) Adaxial epidermis and palisade mesophyll in *in vivo* leaf. (**d**) Adaxial epidermis and palisade mesophyll in *in vitro* leaf. The bottom tab shows that the mucilaginous content of the idioblast pressed the phenolic-containing vacuole to the cell wall; the top tab shows that the mucilage occupied almost the entire cell. (**e**,**f**) Glandular peltate trichomes on the adaxial and abaxial sides of the *in vivo* leaf. (**g**,**h**) Glandular peltate trichomes on the adaxial and abaxial sides of the *in vivo* leaf. AbE—abaxial epidermis, AdE—adaxial epidermis, BS—bundle sheath, Id—mucilage-filled idioblasts; PM—palisade mesophyll, PT—peltate multicellular trichome, SM—spongy mesophyll, St—stomata, UT—unicellular trichome, VB—vessel bundle. The single arrow indicates intravacuolar phenolics; double arrows indicate the outermost epidermal cell wall. ▼—calcium oxalate druse.

**Figure 5 plants-12-04048-f005:**
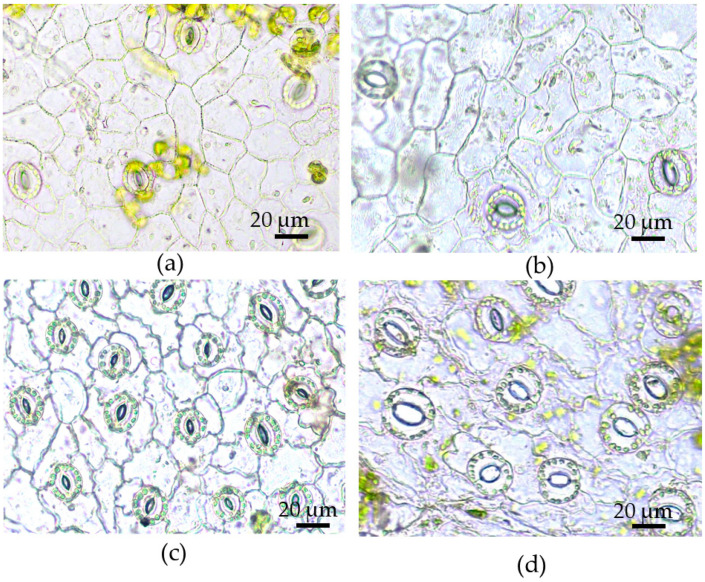
Morphology of stomata on the adaxial (**a**,**b**) and abaxial (**c**,**d**) sides of leaves of buckwheat plants grown *in vivo* (**a**,**c**) and *in vitro* (**b**,**d**).

**Figure 6 plants-12-04048-f006:**
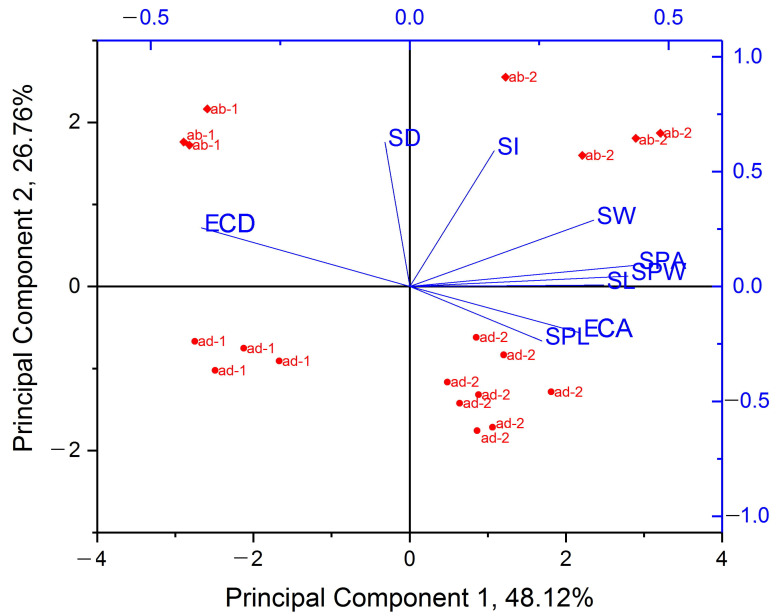
Principal component analysis of epidermal and stomatal characteristics of the adaxial (ab) and abaxial (ad) sides of leaves of Tartary buckwheat plants grown *in vivo* (1) and *in vitro* (2). ECA—epidermal cell area, ECD—epidermal cell density, SD—stomatal density, SI—stomatal index, SW—stomatal width, SL—stomatal length, SPA—stomatal pore area, SPL—stomatal pore length, SPW—stomatal pore width.

**Figure 7 plants-12-04048-f007:**
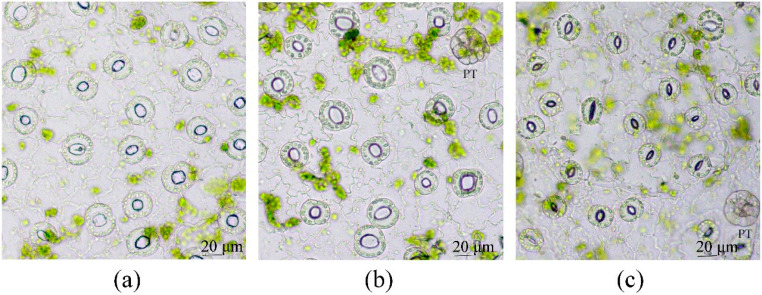
The response of stomata of the abaxial epidermis of *in vitro* buckwheat plants to darkness: (**a**) control, (**b**) 2nd day of culture in darkness, (**c**) 5th day of culture in darkness. PT—peltate trichoma.

**Figure 8 plants-12-04048-f008:**
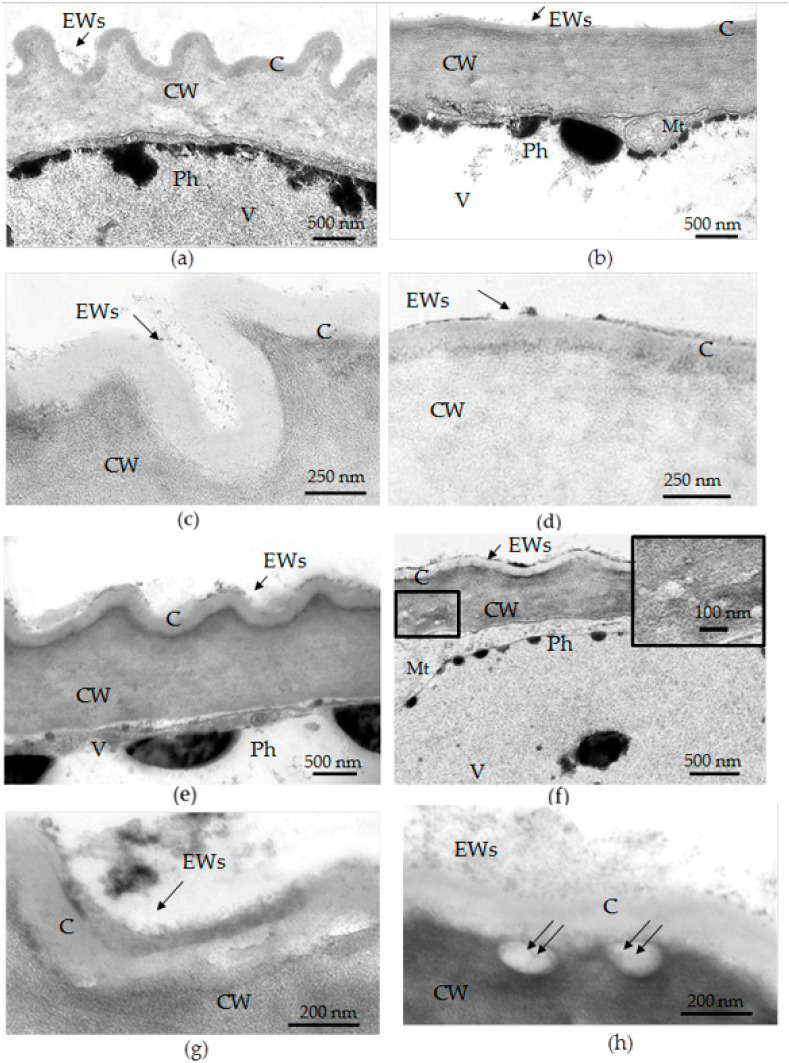
Transmission electron microscopy images of epidermal cell walls and cuticles in leaves of *F. tataricum* grown *in vivo* and *in vitro*. (**a**–**d**) Leaves *in vivo*: (**a**) the external cell wall and cuticle of an adaxial epidermal cell, (**b**) the external cell wall and cuticle of an abaxial epidermal cell, (**c**) the cuticular ridges on the surface of an adaxial epidermal cell covered with epicuticular waxes, (**d**) the cuticle of abaxial epidermis with surficial layer of epicuticular waxes. (**e**–**h**) Leaves *in vitro*: (**e**) the external cell wall and cuticle of an adaxial epidermal cell, (**f**) the outer cell wall and cuticle of an abaxial epidermis—the cell wall with “bubbles” or caverns and irregularities of cellulose microfibrils (on the tab) and cuticle covered with epicuticular waxes, (**g**) the cuticle of adaxial epidermis with clumps of waxes and possible polysaccharides on its surface—the discontinuity of cuticle is marked by double arrows, (**h**) the cuticle of abaxial epidermis covered by an amorphous network of epicuticular waxes and possible polysaccharides. Epicuticular waxes marked by single arrow; the caverns between cuticle and cell wall are marked by double arrows. EWs—epicuticular waxes, C—cuticle, CW—cell wall, Mt—mitochondria, Ph—phenolics, V—vacuole.

**Figure 9 plants-12-04048-f009:**
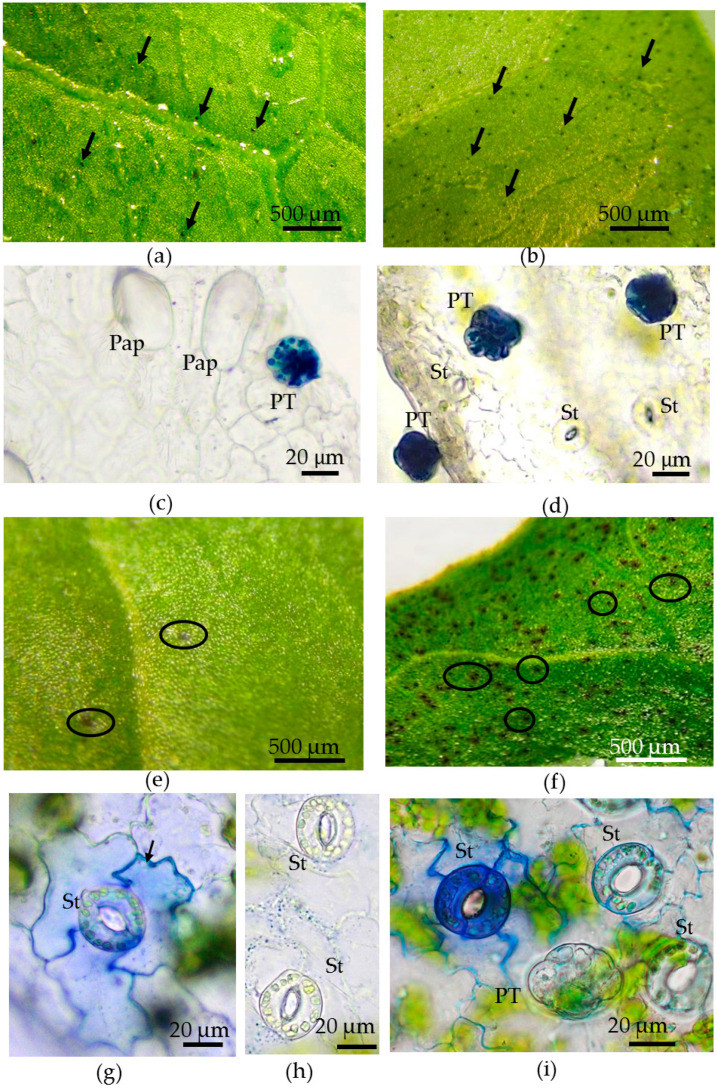
Examination of cuticle defects by TB test on the adaxial and abaxial surfaces of leaves of Tartary buckwheat plants grown *in vivo* and *in vitro*: (**a**–**d**) *In vivo* leaf, 30 min of TB staining: (**a**) sparse, colored structures, marked by arrows, on the adaxial leaf surface; (**b**) numerous colored structures, marked by arrows, on the abaxial leaf surface; (**c**) stained peltate trichomes and unstained papillaes on the adaxial leaf surface; (**d**) stained peltate trichomes and unstained stomata on the abaxial leaf surface. (**e**–**i**) *In vitro* leaf, 7 min of TB staining: (**e**) single purple spots, highlighted by a circle, on the adaxial surface of the leaf; (**f**) multiple colored spots, highlighted by a circle, on the abaxial surface of the leaf; (**g**) adaxial epidermis, single spots on the leaf—staining of stomatal guard cells and cell walls of adjacent pavement cells; (**h**) unstained stomata and epidermal cells on the adaxial surface of the leaf; (**i**) different intensity of staining of stomatal guard cells and cell walls of pavement cells on the abaxial surface of the leaf—no staining of peltate trichomes. Examination of cuticle defects by TB test on the adaxial and abaxial surfaces of leaves of Tartary buckwheat plants grown *in vivo* and *in vitro* (continued). (**j**) *In vitro* leaf, 10 min of TB staining, increase in the number of stained spots on the abaxial surface of the leaf. (**k**) *In vitro* leaf, 10 min of TB staining, the larger imagination of the abaxial side of the same leaf, shown in Figure 9j. (**l**) *In vitro* leaf (epidermal strips), 10 min of TB staining, irregular coloring of stomata and epidermal cells on the abaxial surface of the leaf and unstained peltate trichome. (**m**–**p**) *In vitro* leaf, 30 min of TB staining, the different intensity and character of coloring of the adaxial (**m**,**n**) and abaxial (**o**,**p**) leaf surface—intensely colored cells of the abaxial surface (**p**) “shine through” the uncolored adaxial epidermis (**n**), on which individual tiny colored spots marked with arrows (**p**) are visible. (**q**) *In vitro* leaf (adaxial epidermis), 30 min of TB staining—peltate trichomes are colored. (**r**) *In vitro* leaf (abaxial epidermis), 30 min of TB staining—uneven intensity of coloring of stomata and surrounding cells is shown, stained peltate trichomes are indicated by an arrow. (**s**) Intense staining of guard cells of stomata (abaxial epidermis), peltate trichomes, and small colored structures inside pavement cells. (**t**) Abaxial epidermal stripe, staining of stomata and cell walls in pavement cells, marked by arrow.

**Figure 10 plants-12-04048-f010:**
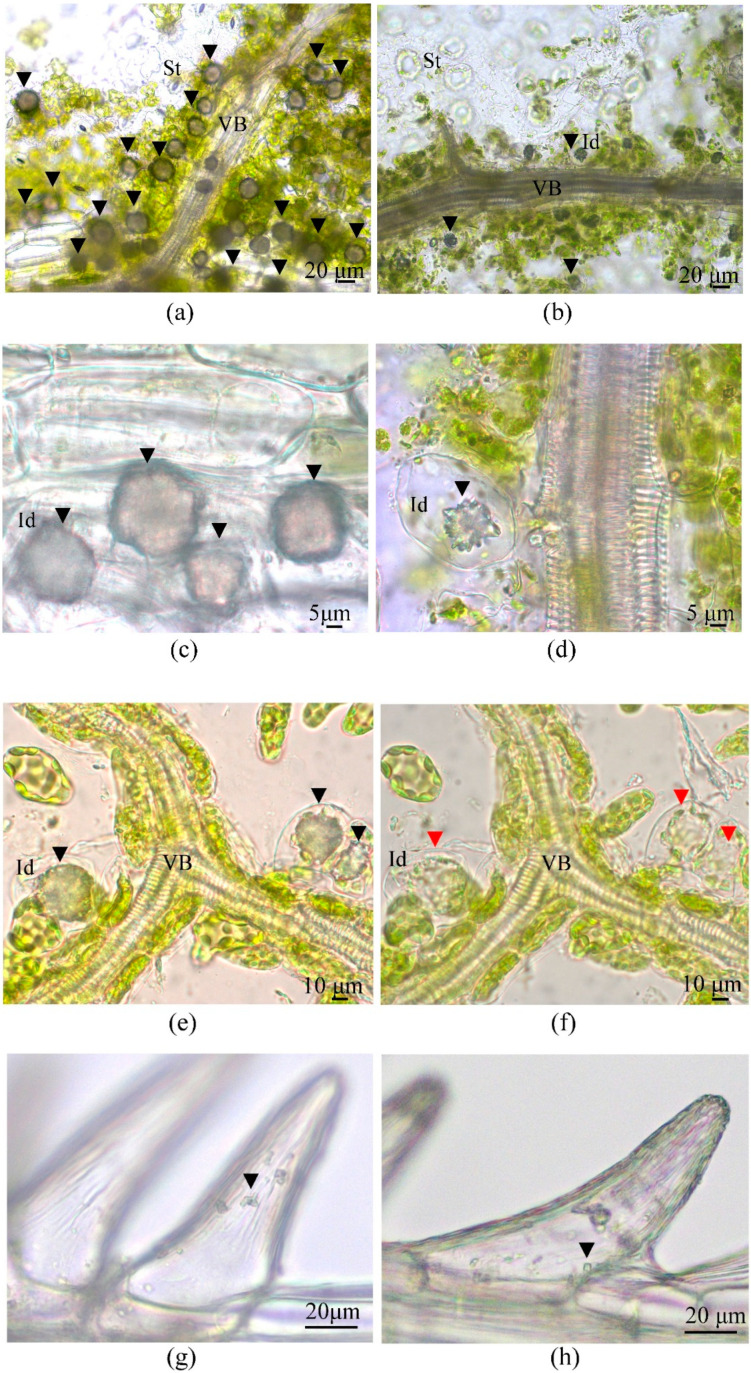
Distribution of calcium oxalate (CaOx) druses in Tartary buckwheat leaves grown *in vivo* (**a**,**c**,**e**–**h**) and *in vitro* (**b**,**d**). Vital vein preparations (**a**–**d**) of outdoor (*in vivo*) plant leaves (**a**,**c**) and leaves grown *in vitro* (**b**,**d**). Identification of CaOx by its dissolution (**e**,**f**) before (**e**) and after (**f**) 30 min of Na-EDTA addition. Deposition of cuboidal CaOx druses in unicellular trichomes of *in vivo* (**g**,**h**) leaves. Id—idioblast, ▼—CaOx druse, ▼—dissolution of CaOx druse, VB—vessel bundle, St—stomata.

**Figure 11 plants-12-04048-f011:**
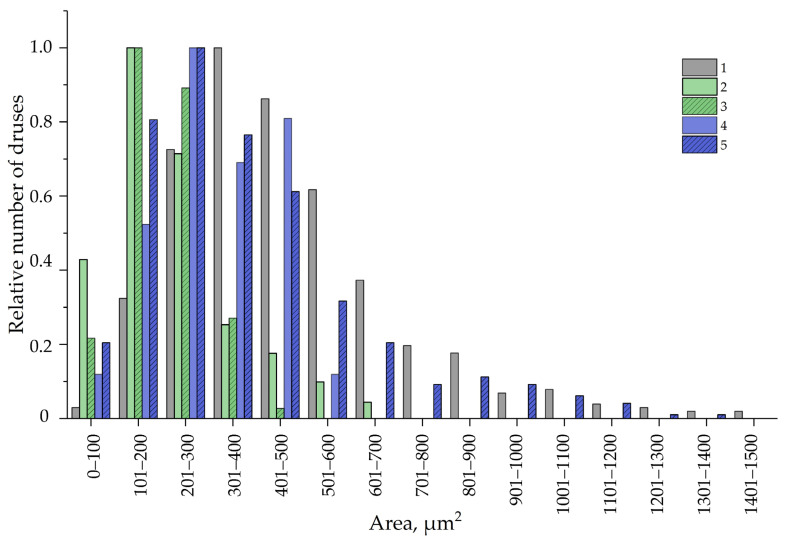
The distribution of CaOx druses according to their area in leaves of Tartary buckwheat plants grown *in vivo* for 30 days, and *in vitro* both on MS medium with standard CaCl_2_ concentration (2- for 30 days, 3- for 45 days), and on MS medium with double concentration of CaCl_2_ (4- for 30 days, 5- for 45 days). The distribution was normalized from data containing 465 measurements of druses found in leaves of outdoor plants, 247 and 89 measurements of druses found in leaves of plants grown 30 days and 45 days on MS medium with standard Ca, and 137 and 424 measurements of druses found in leaves of plants grown on double Ca MS medium for 30 days and 45 days, respectively.

**Figure 12 plants-12-04048-f012:**
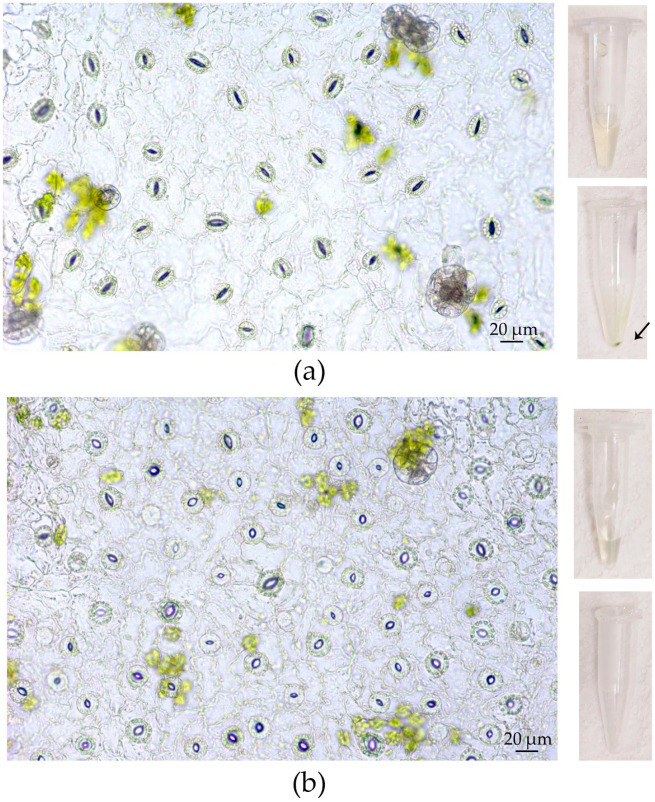
Effect of “wet chamber” conditions on stomatal morphology and purity of apoplastic washing fluid (AWF) obtained by VIC procedure at 900× *g* from leaves of Tartary buckwheat plants grown in vegetation site (*in vivo*). (**a**) Control plants: closed and semi-closed stomata of abaxial epidermis, yellowish color of AWF, precipitate (indicated by an arrow) formed after centrifugation of AWF at 10,000× *g*. (**b**) Buckwheat plants kept in “wet chamber” conditions for 2.5 h: open stoma of abaxial epidermis, almost colorless AWF, absence of precipitate after AWF centrifugation at 10,000× *g*. PT—peltate trichoma.

**Figure 13 plants-12-04048-f013:**
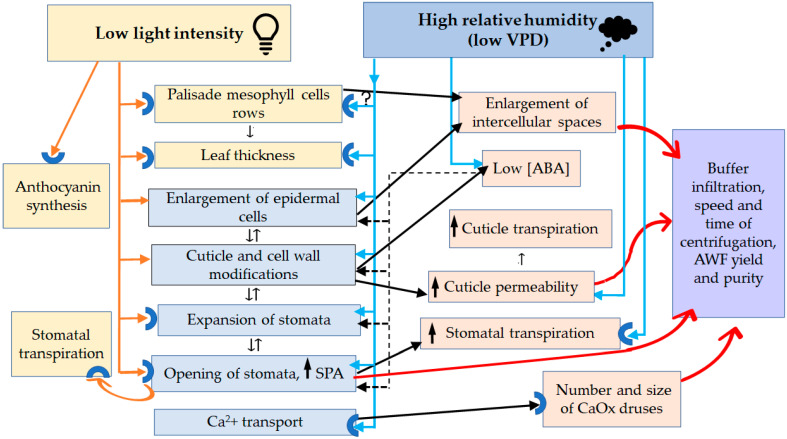
The impact of low lighting and high humidity on the spectrum of adaptive physiological, morphological, and anatomical changes in the leaves of Tartary buckwheat cultured *in vitro*. A question mark (?) indicates that there is no clear experimental data on whether high RH affects the number of rows of palisade mesophyll cells.

**Table 1 plants-12-04048-t001:** Effect of centrifugal force on AWF collection efficiency and apoplastic protein content from leaves of Tartary buckwheat plants grown *in vivo* and *in vitro*.

Culture Condition	Centrifugal Force, g	AWF Yield, µL/g FW	AWF Protein Content, µg/g FW
*In vivo*	200	165.47 ± 37.52 Aa	9.59 ± 2.47 Aa
	600	288.66 ± 13.15 Ab	19.55 ± 1.48 Ab
	900	302.07 ± 29.96 Ab	39.47 ± 4.79 Ac
	1500	455.58 ± 42.47 Ac	57.46 ± 12.77 Acd
	3000	581.72 ± 46.25 Ac	87.18 ± 15.38 Ad
*In vitro*	200	228.04 ± 32.69 Aa	13.46 ± 2.46 Aa
	600	344.41 ± 15.09 Bb	36.70 ± 2.81 Bb
	900	402.23 ± 26.66 Bc	50.34 ± 4.56 Ac
	1500	477.92 ± 45.91 Acd	54.64 ± 9.79 Acd
	3000	578.51 ± 69.58 Ad	88.12 ± 13.21 Ad

Significant differences between samples (*p* ≤ 0.05) are marked with different letters. Uppercase letters indicate the difference between the variants *in vitro* and *in vivo* at the same centrifugal force; lowercase letters indicate the effect of centrifugal force within the same object (*in vitro* or *in vivo*).

**Table 2 plants-12-04048-t002:** Characteristics of leaves of Tartary buckwheat grown *in vivo* and *in vitro*.

Traits	The Conditions of Plant Growth		PPI
	*In Vivo*	*In Vitro*	
Leaf dry mass, %	15.81 ± 0.28 b	12.24 ± 0.67 a	0.23
Leaf area, cm^2^	8.62 ± 0.49 b	0.60 ± 0.05 a	0.93
Leaf length, cm	2.94 ± 0.09 b	0.92 ± 0.41 a	0.69
Leaf width, cm	3.46 ± 0.12 b	0.94 ± 0.05 a	0.73
Leaf blade thickness, µm	153.87 ± 3.91 b	132.12 ± 3.05 a	0.14

Data are means ± standard error. Means followed by different letters are significantly different by *t*-test (*p* ≤ 0.05). Different letters indicate significant differences between plants *in vivo* and *in vitro*. PPI, phenotypic plasticity index.

**Table 3 plants-12-04048-t003:** Characteristics of adaxial and abaxial sides of leaves of Tartary buckwheat plants grown *in vivo* and *in vitro*.

Traits	The Conditions of Plant Growth		PPI
	*In Vivo*	*In Vitro*	
**Epidermal cell area, µm^2^**			
adaxial	519.69 ± 32.97 Aa	1201.63 ± 43.97 Bb	0.57
abaxial	488.30 ± 30.62 Aa	959.26 ± 34.43 Ab	0.49
**Epidermal cell density, mm^−2^**			
adaxial	1392.47 ± 57.53 Ab	924.71 ± 41.92 Aa	0.34
abaxial	1876.53±175.71 Bb	1021.81±41.38 Aa	0.46
**Stomatal density, mm^−2^**			
adaxial	99.57 ± 3.56 Ab	76.83 ± 4.79 Aa	0.23
abaxial	343.60 ± 13.75 Bb	288.10 ± 11.94 Ba	0.16
**Stomatal index, %**			
adaxial	6.70 ± 0.42 Aa	7.50 ± 0.61 Aa	0.11
abaxial	15.80 ± 1.89 Ba	22.20 ± 0.38 Bb	0.29
**Stomatal length, µm**			
adaxial	23.18 ± 0.24 Ba	24.75 ± 0.30 Ab	0.06
abaxial	22.02 ± 0.34 Aa	26.15 ± 0.47 Bb	0.16
**Stomatal width, µm**			
adaxial	20.11 ± 0.25 Aa	22.38 ± 0.44 Ab	0.10
abaxial	19.94 ± 0.29 Aa	26.67 ± 0.64 Bb	0.25
**Stomatal pore length, µm**			
adaxial	12.14 ± 0.18 Ba	12.61 ± 0.28 Aa	0.04
abaxial	10.63 ± 0.22 Aa	12.42 ± 0.41 Ab	0.15
**Stomatal pore width, µm**			
adaxial	5.07 ± 0.18 Aa	7.16 ± 0.28 Ab	0.29
abaxial	5.39 ± 0.20 Aa	9.38 ± 0.24 Bb	0.43
**Stomatal pore area, µm^2^**			
adaxial	31.89 ± 1.04 Aa	58.98 ± 2.75 Ab	0.46
abaxial	29.59 ± 1.00 Aa	82.41 ± 3.85 Bb	0.64

Data are means ± standard error. Means followed by different letters are significantly different by *t*-test (*p* ≤ 0.05). Uppercase letters indicate differences between values for adaxial and abaxial epidermis within *in vivo* or *in vitro* leaves; lowercase letters indicate differences between plants *in vivo* and *in vitro*. PPI, phenotypic plasticity index.

**Table 4 plants-12-04048-t004:** The thickness of the cell walls and cuticles of epidermal cells in leaves of Tartary buckwheat grown *in vivo* and *in vitro*.

Traits	The Conditions of Plant Growth		PPI
	*In Vivo*	*In Vitro*	
Cell wall thickness, nm			
adaxial epidermis	1095.50 ± 27.71 Ba	1038.16 ± 26.20 Ba	0.05
abaxial epidermis	879.42 ± 14.70 Ab	594.71 ± 14.91 Aa	0.32
Cuticle thickness, nm			
adaxial epidermis	212.76 ± 4.95 Bb	179.86 ± 4.15 Ba	0.16
abaxial epidermis	111.14 ± 2.21 Ab	150.56 ± 6.52 Aa	0.26

Means followed by different letters vary significantly by *t*-test (*p* ≤ 0.05). Uppercase letters mean the difference between the adaxial and abaxial sides of leaves of the same variant (*in vivo*- or *in vitro*-grown plants); lowercase letters mean the difference in the same row between plants grown *in vivo* and *in vitro*. PPI, phenotypic plasticity index.

**Table 5 plants-12-04048-t005:** Calcium oxalate druse parameters in leaves of Tartary buckwheat plants grown *in vivo* and *in vitro*.

Traits	The Conditions of Plant Growth		PPI
	*In Vivo*	*In Vitro*	
Druse density, per mm^2^ of tissue	217.24 ± 14.17 b	75.46 ± 5.18 a	0.65
Area of calcium oxalate druses, µm^2^	469.16 ± 10.93 b	218.00 ± 8.49 a	0.54

Data are means ± standard error. Means followed by different letters are significantly different by *t*-test (*p* ≤ 0.05). Different letters indicate significant differences between plants *in vivo* and *in vitro*. PPI, phenotypic plasticity index.

## Data Availability

Data is contained within the article or Appendix A.

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
