# Peer review of "The Effect of Leaf Plasticity on the Isolation of Apoplastic Fluid from Leaves of Tartary Buckwheat Plants Grown In Vivo and In Vitro"

_plants, 2023, doi:10.3390/plants12234048_

Round 1
Reviewer 1 Report
Comments and Suggestions for Authors For the manuscript (ID plants-2691698) “The effect of leaf plasticity on the isolationof apoplastic fluid from leaves of tartary buckwheat plants grown in vivo and in vitro”
written by Natalya I. Rumyantseva, Alfia I. Valieva, Yulia A. Kostyukova,
Marina V. Ageeva
The aim of this research was to elaborate an optimal procedure for apoplast washing fluid (AWF) isolation from the leaves of Fagopyrum tataricum basing on the authors’ data about the broad range of leaf micromorphological traits in plants grown in vivo and in vitro. The MS is potentially of interest, but it can’t be recommended for publication because requires major revision to improve the perception of the experimental data presented and their analysis.
Comments Present description of the plant material is utterly inadequate for a value of the obtained
results. It is unknown the state of plants grown outdoors, what leaves were taken to study,
weather conditions, explant description, what does it mean “For all experiments,
in vitro- and in vivo- grown plants were 30-day old” etc. Preferably to decrease the volume of the “Introduction” by reducing a common
well-known information and information not directly related to the subject of research. It is necessary to edit the text avoiding the repetition of material in different sections
and unsuccessful expressions, such as “Figure 4. Transverse histological sections of
Tartary buckwheat leaves…, the anatomical structure of the adaxial epidermis”
… lines 484, 485 etc. In the Discussion, according to the aim of research clearly state the authors' ideas on
the optimal procedure for AWF isolation from leaves and practical recommendations
based on the data obtained. Scheme 1. Conclusion and future directions as presented
is unacceptable. In general, the text is overloaded with data on micromorpholgical traits of leaves
in vivo and in vitro that could be presented more briefly; the details may be transferred
to the supplement, as well as the section 3.6. Estimation of the phenotypic plasticity of
Fagopyrum tataricum leaves in response to in vitro conditions. I suggest that authors
consider dividing the MS into two papers. Comments on the Quality of English Language
.
Author Response
Dear reviewer,
We thank you for your review, valuable recommendation, questions, and constructive remarks. We have done our best to fix the flaws in the manuscript.
Let me answer your comments and questions.
Present description of the plant material is utterly inadequate for a value of the obtained results. It is unknown the state of plants grown outdoors, what leaves were taken to study, weather conditions, explant description, what does it mean “For all experiments, in vitro- and in vivo- grown plants were 30-day old” etc.
The section Matherials and methods was supplemented by the description on the state of the plants grown outdoors; it was also described which leaves were taken for study, and weather conditions (Table S1).
Explant description. We used tissue pieces approximately 3 × 5 mm from the area of lateral leaf veins for both TEM and histology (Figure S1, black squares). From the same areas, the epidermal strips for morphometry of stomata and pavement cells were made (the strip sizes were no more than 2x4 mm). CaOx druse study was carried out on preparations of epidermal peels with vein of first order and adjacent mesophyll cells (Figure S1, red squares). First, part of the vein with adjacent spongy mesophyll cells and the abaxial epidermis were separated with a needle, and then taken them off with forceps as a strip. For CaOx druse study, epidermal strips were taken off only from abaxial side of the leaf (Figure S1).
What does it mean “For all experiments,in vitro- and in vivo- grown plants were 30-day old” etc
In this work we used outdoor plants of 4-5 wk old and in vitro plants of 28-30 days; the changes in description have been made to the text.
Preferably to decrease the volume of the “Introduction” by reducing a common well-known information and information not directly related to the subject of research.
We have done this.
It is necessary to edit the text avoiding the repetition of material in different sections and unsuccessful expressions, such as “Figure 4. Transverse histological sections of Tartary buckwheat leaves…, the anatomical structure of the adaxial epidermis”
… lines 484, 485 etc.
We thoroughly edited the text and diminished its volume and exclude the repetition, unsuccessful expression, and unnecessary details as much as possible.
In the Discussion, according to the aim of research clearly state the authors' ideas on the optimal procedure for AWF isolation from leaves and practical recommendations based on the data obtained.
We have done this.
Scheme 1. Conclusion and future directions as presented is unacceptable.
It was the mistake of formatting when the legend of Scheme was changed by the title of the next part of manuscript (Conclusion and further perspectives). Correction was made.
In general, the text is overloaded with data on micromorpholgical traits of leaves in vivo and in vitro that could be presented more briefly; the details may be transferred to the supplement, as well as the section 3.6. Estimation of the phenotypic plasticity of Fagopyrum tataricum leaves in response to in vitro conditions.
The text of manuscript was shortened. The Figure 12 was transferred to the Supplement.
I suggest that authors consider dividing the MS into two papers.
I realize that the value of experimental data is maybe enough for two manuscripts, but these data are interrelated and complement each other, so would not like to divide them.
Sincerely yours,
Natalya Rumyantseva

Reviewer 2 Report
Comments and Suggestions for Authors
Dear Author
Regarding to the manuscript ID: plants-2691698
Title:
The Effect of Leaf Plasticity on the Isolation of Apoplastic Fluid from Leaves of Tartary Buckwheat Plants Grown and in vivo and in vitro
. The main aim of the study is addressed and studied an important point and obtained good results-
.The topic is original and relevant to the field- .The methods are adequate and written in a good way-
.The tables and figures are sufficient and provided in a clear form-
.The current information will be necessary and important for the researches who work in this area-
The current study supplied important information when compared the in vivo and in vitro leaf plasticity of tartary buckwheat plants
: Some specific comments should be taken in consideration to improve the quality of the manuscript
- Abstract must be rewritten with more details
- The introduction is long and need to be shortening.
- The use of personal pronouns such as "we, our; i, etc." is not recommended in scientific language. Please revise the manuscript and avoid using it as much as it is possible. Please revise and correct all over the manuscript
- There are some writing mistakes, so the manuscript should be revised carefully
- The sentences should not be started with abbreviation; it is better to start with complete form, please correct all over the manuscript.
- The conclusion must be added and include result details.
- There are several mistakes in the references, please revise carefully and rewrite
according the journal format as well as the title of the cited articles with an uppercase letter in the initial only for its first word, please revise and correct all the reference list.
- There are several comments provided in the attached manuscript should be taken in consideration.
Responding to these comments would improve the overall quality and readability of the manuscript.
Best regards

Minor editing of English language required
Author Response
Dear reviewer,
We express our sincere gratitude for your interest in our work, and for your valuable remarks and recommendations.
Let me answer your comments and recommendations.
- Abstract must be rewritten with more details
The abstract was rewritten, and more details were added.
- The use of personal pronouns such as "we, our; i, etc." is not recommended in scientific language. Please revise the manuscript and avoid using it as much as it is possible. Please revise and correct all over the manuscript
We carefully revised the manuscript and diminished the overage of “we”, “us”, “our” et cet.
- There are some writing mistakes, so the manuscript should be revised carefully
It has been completed; manuscript was revised.
- The introduction is long and need to be shortening.
The introduction was shortened.
- Do not start sentence with abbreviation; it is better to start with complete form, please correct all over the manuscript.
Your remark has been taken in consideration, and corrections were made.
- The conclusion must be added and include result details.
It has been done.
- There are several mistakes in the references, please revise carefully and rewrite according the journal format as well as the title of the cited articles with an uppercase letter in the initial only for its first word, please revise and correct all the reference list.
The list of references has been corrected according to the journal format.
- There are several comments provided in the attached manuscript should be taken in consideration.
All comments have been taken in consideration. Thank you.
The values in the Table1 were recalculated, since the reviewer had doubts about their statistical validity. Nothing has changed much in the conclusions and results, but the quality of statistics has improved. I admit that the statistical processing in the Table 1 was carried out incorrectly.
We have checked the writing of the terms “Parafilm” and “Miracloth”. Both of them are capitalized in articles, probably because they are registered trademarks. Therefore, we have left this writing in the article for now.
Best regards,
Natalya Rumyantseva
Round 2
Reviewer 1 Report
Comments and Suggestions for Authors
Authors have made essential corrections. MS can be published.
Comments on the Quality of English Language The quality of English is satisfactory.